# aPKC-mediated displacement and actomyosin-mediated retention polarize Miranda in *Drosophila* neuroblasts

**Matthew Robert Hannaford, Anne Ramat, Nicolas Loyer, Jens Januschke***

Cell and Developmental Biology, School of Life Sciences, University of Dundee, Dundee, United Kingdom

**Abstract** Cell fate assignment in the nervous system of vertebrates and invertebrates often hinges on the unequal distribution of molecules during progenitor cell division. We address asymmetric fate determinant localization in the developing *Drosophila* nervous system, specifically the control of the polarized distribution of the cell fate adapter protein Miranda. We reveal a stepwise polarization of Miranda in larval neuroblasts and find that Miranda's dynamics and cortical association are differently regulated between interphase and mitosis. In interphase, Miranda binds to the plasma membrane. Then, before nuclear envelope breakdown, Miranda is phosphorylated by aPKC and displaced into the cytoplasm. This clearance is necessary for the subsequent establishment of asymmetric Miranda localization. After nuclear envelope breakdown, actomyosin activity is required to maintain Miranda asymmetry. Therefore, phosphorylation by aPKC and differential binding to the actomyosin network are required at distinct phases of the cell cycle to polarize fate determinant localization in neuroblasts.
DOI: https://doi.org/10.7554/eLife.29939.001

*For correspondence:
j.januschke@dundee.ac.uk

Competing interests: The authors declare that no competing interests exist.

## Introduction

The development of the central nervous system depends on asymmetric cell divisions for the balanced production of progenitor and differentiating cells. During vertebrate and invertebrate neurogenesis, cell fates can be established through the asymmetric inheritance of cortical domains or fate determinants during asymmetric division of progenitor cells (*Alexandre et al., 2010*; *Doe, 2008*; *Knoblich, 2008*; *Marthiens and ffrench-Constant, 2009*).

A vital step in asymmetric cell division is the establishment of a polarity axis. Asymmetrically dividing *Drosophila* neuroblasts (NBs) establish an axis of polarity at the onset of mitosis and, as in many other polarized cells, this depends on the activity of the Par protein complex (*Goldstein and Macara, 2007*). As NBs enter prophase, the Par complex, comprising Par3/Bazooka (Baz), aPKC and Par-6, assembles at the apical NB pole, and this drives the localization of fate determinants to the opposite (basal) NB pole, thus establishing the apico–basal polarity axis (*Betschinger et al., 2003*; *Homem and Knoblich, 2012*; *Petronczki and Knoblich, 2001*; *Prehoda, 2009*; *Rolls et al., 2003*; *Wodarz et al., 2000*; *1999*).

Upon NB division, the basally-localized fate determinants segregate to the daughter cell, which then commits to differentiation. Two adapter proteins localize the fate determinants to the basal NB cortex in mitosis: Partner of Numb (Pon), localizes the Notch signaling regulator Numb (*Lu et al., 1998*; *Uemura et al., 1989*), and Miranda (Mira), (*Ikeshima-Kataoka et al., 1997*; *Shen et al., 1997*) localizes fate determinants including the homeobox transcription factor Prospero (Pros) and the translational repressor Brat (*Betschinger et al., 2006*; *Ikeshima-Kataoka et al., 1997*; *Lee et al., 2006*). In the absence of Mira, fate determination is impaired and tumor-like growth can occur in larval NB lineages (*Caussinus and Gonzalez, 2005*; *Ikeshima-Kataoka et al., 1997*).

Intriguingly, Mira is uniformly cortical in interphase larval brain NBs (*Sousa-Nunes et al., 2009*). In embryonic NBs, Mira and its cargo Pros co-localize, at the interphase cortex (*Spana and Doe, 1995*). The cortical localization of Pros seems to depend on Mira, since in interphase *mira* mutant NBs, Pros is found in the NB nucleus (*Matsuzaki et al., 1998*). Given that the levels of nuclear Pros in NBs are important for the regulation of entry and exit from quiescence (*Lai and Doe, 2014*), Mira localization and its regulation are likely to be relevant for regulating nuclear Pros levels in interphase NBs, but how this might be achieved is unknown.

How the asymmetric localization of Miranda in mitotic NBs is achieved is also a long-standing question, however, the mechanism is still not fully understood. In embryonic NBs, Mira localization requires actin (*Shen et al., 1998*) and myosin activity, since mutation in the myosin regulatory light chain *spaghetti squash* (*sqh, Barros et al., 2003*) or the Myosin VI *jaguar* (*Petritsch et al., 2003*) lead to Mira localization defects. Moreover, Mira does not achieve a polarized distribution in embryos into which the Rho kinase (ROCK) inhibitor Y-27632 was injected. ROCK can regulate Myosin activity by phosphorylating the light chain of Myosin (*Amano et al., 1996*). Furthermore, the effect of pharmacological ROCK inhibition on Mira in embryonic NBs could be rescued by the expression of a phosphomimetic version of Myosin's regulatory light chain, Spaghetti Squash (called SqhEE, *Winter et al., 2001*), which led to the idea that Myosin II might play a critical role in Mira localization and that aPKC affects Mira localization indirectly through regulating Myosin II (*Barros et al., 2003*).

However, Y-27632 can inhibit aPKC directly, and Mira is a substrate of aPKC (*Atwood and Prehoda, 2009*; *Wirtz-Peitz et al., 2008*). In fact many aPKC substrates, including Numb and Mira, contain a basic and hydrophobic (BH) motif that can be phosphorylated by aPKC. When phosphorylated, the substrates can no longer directly bind phospholipids of the plasma membrane (PM), (*Bailey and Prehoda, 2015*; *Dong et al., 2015*; *Smith et al., 2007*). Thus, asymmetric Mira localization in mitotic NBs can, in principle, be explained by keeping the activity of aPKC restricted to the apical pole. According to this model, Mira retention at the cortex is primarily mediated through direct interaction with the PM mediated by its BH motif. Therefore, although the contribution of actin was revealed in pharmacological and genetic experiments, it remains unclear how actin contributes to fate determinant localization.

To understand the regulation of Mira localization throughout the NB cell cycle, we set out to determine differences and similarities in the parameters of Mira binding in interphase and mitosis, and analysed the transition between the two different localization patterns. Mira has been shown to be able to localize to microtubules (*Mollinari et al., 2002*) and directly bind actin (*Sousa-Nunes et al., 2009*) and phospholipids of the PM (*Bailey and Prehoda, 2015*). Therefore, we re-examined the role of the cytoskeleton and PM interaction for Mira localization. We reveal that Mira uses two modes to interact with the cortex: in interphase, to retain Mira uniformly at the cortex direct interaction of Mira's BH motif with phospholipids of the PM are necessary and likely sufficient. This interaction is inhibited by aPKC-dependent phosphorylation of the BH motif at prophase. After nuclear envelope breakdown Mira requires BH motif and actomyosin-dependent processes for asymmetric retention at the cortex. Therefore, we propose that Mira binds to the PM in interphase and to the actomyosin cortex in mitosis, both of which appear BH motif dependent.

## Results

### Uniform Miranda is cleared from the cortex during prophase and reappears asymmetrically localized after nuclear envelope breakdown

We confirmed that Mira localizes uniformly to the cortex in interphase larval NBs and also find that its cargo Pros localizes to the interphase cortex in a Mira-dependent manner (*Figure 1A*, *Figure 1—figure supplement 1*). These results suggest that Mira regulates the localization of its cargoes throughout the cell cycle. Therefore, we sought to address the regulation of cortical Mira in interphase and mitosis, and the transition between these localizations patterns of Mira.

To monitor the establishment of asymmetric Mira localization, we used a BAC construct in which Mira was tagged with mCherry at its C-terminus (see also *Figure 1—figure supplement 2*). This tagged Mira recapitulated uniform cortical localization in interphase (*Figure 1B*, −33 to NEB) and polarized localization to the basal pole in mitosis (*Figure 1B*, +4), showing a 2.5-fold increase in

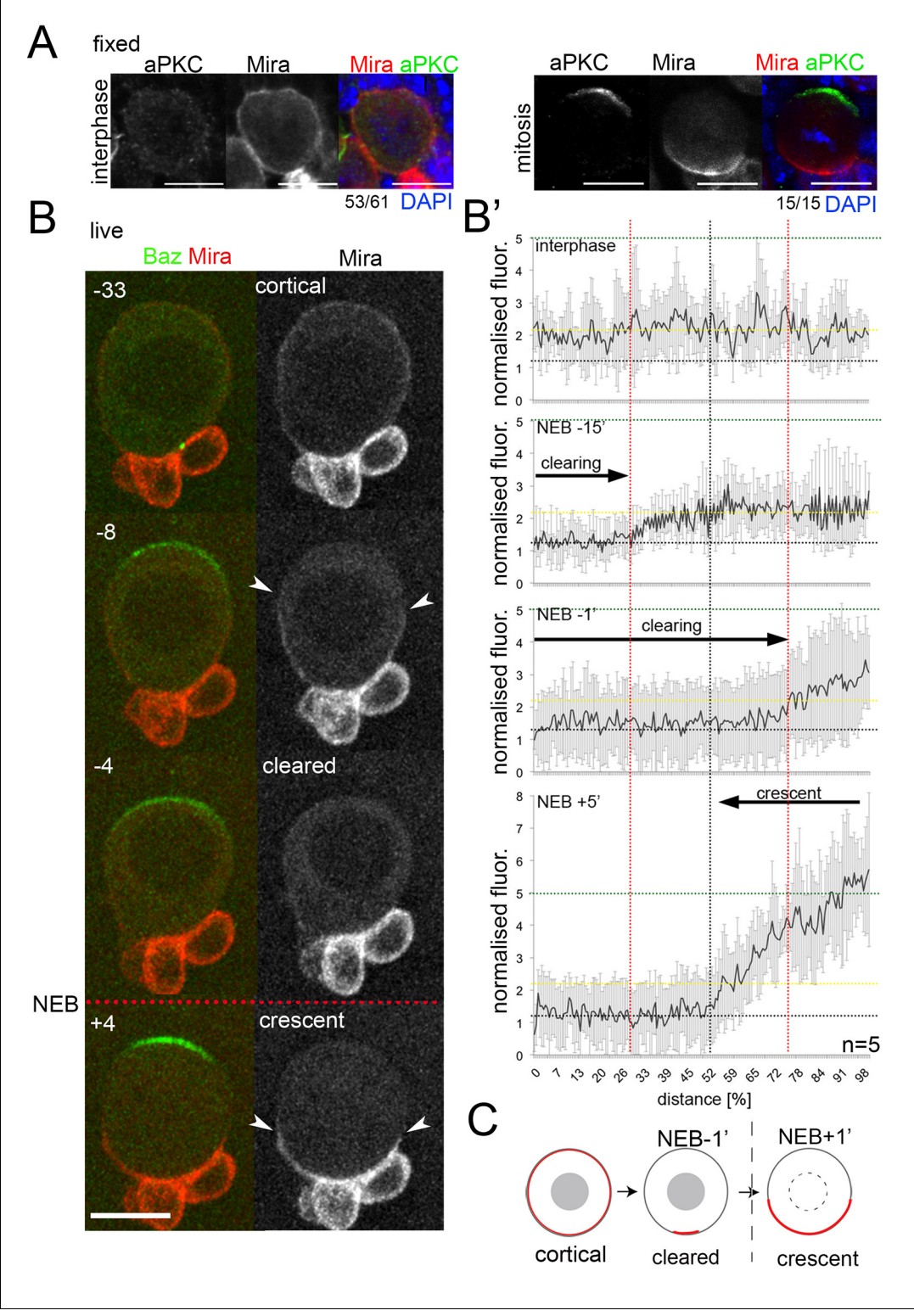

**Figure 1.** Miranda is cleared from the cortex before localizing in a basal crescent in mitosis. (A) Larval brain NBs fixed and stained as labeled at the indicated cell cycle stage. (B) Selected frames from *Video 1*. NB in primary cell culture expressing Baz::GFP (green) and Mira::mCherry (red) in the transition from interphase to mitosis. Arrowheads point at Mira being cleared (−8) and at basal Mira crescent (+4). (B') Quantification of cortical Mira:: mCherry signal plotting the fluorescence intensities from the apical to the basal pole computationally straightening (*Kocsis et al., 1991*) the cortices of five NBs against the distance in percent. Fluorescence was

*Figure 1 continued on next page*

*Figure 1 continued*

background subtracted and normalized to background subtracted cytoplasmic signal (1, dotted line). Cortical signal (yellow dotted line) and signal after NEB (green dotted line). Error bars, standard deviation. (C) Schematic of Mira localization. *BAC{mira::mcherry-MS2}* was the source of Mira::mCherry. Scale bar 10 μm. Time stamp: minutes.

DOI: https://doi.org/10.7554/eLife.29939.002

The following figure supplements are available for figure 1:

**Figure supplement 1.** Uniform cortical Prospero depends on Miranda in interphase larval NBs.

DOI: https://doi.org/10.7554/eLife.29939.003

**Figure supplement 2.** *BAC{mira::mcherry-MS2}* rescues embryonic lethality of the loss of function allele mira[L44] over the deficiency DF(3R)ora[I9].

DOI: https://doi.org/10.7554/eLife.29939.004

**Figure supplement 3.** Cortical Mira can be detected by antibody staining, in UAS-GFP-Mira overexpressing NBs and upon colcemid treatment, but not in interphase mira[L44] loss of function clones.

DOI: https://doi.org/10.7554/eLife.29939.005

intensity at the basal cortex (n = 5, *Figure 1B'*). This transition occurred in two distinct steps: First, during prophase, Mira was rapidly excluded from the apical pole, where Baz (*Video 1*) and aPKC (*Video 2*) began to accumulate (*Figure 1B* -8 to NEB, *Figure 1B'* −15). Subsequently, Mira was progressively cleared from most of the rest of the cortex in an apical-to-basal direction (*Figure 1B,B'* -4, −1, respectively to NEB); Second, following NEB, Mira reappeared at the cortex in a basal crescent (*Figure 1B,B'* +4, +5 respectively). We recapitulated these steps using overexpression of GFP::Mira and by antibody staining of endogenous, non-tagged Mira (*Figure 1—figure supplement 3*).

In conclusion, Miranda transitions from a uniformly cortical localization with low intensity levels in interphase, to a basal localization with high intensity levels in metaphase (*Figure 1B,B', C*). These cell-cycle dependent differences in cortical Mira intensities prompted the idea that Mira might use different modes of binding to the cortex in interphase versus mitosis. Therefore, we assayed for potential differences in cortical binding of Mira in interphase versus mitosis to address whether Mira is retained at the cortex primarily by BH motif interaction with the PM, or whether other modes of cortical retention contribute.

## Actomyosin is required for both establishment and maintenance of Miranda crescents

We started by re-examining the role of the actin cytoskeleton by disrupting it with Latrunculin A (LatA). F-actin has been shown to be involved in Mira localization in mitotic embryonic NBs (*Shen et al., 1998*). Therefore, we tested if an intact actin network was also required for Miranda localization in larval NBs. Despite efficient disruption of F-actin (*Figure 2—figure supplement 1*) causing cytokinesis failure (*Figure 2A*, 3:02, related to *Video 3*), LatA treatment did not affect the uniform interphase cortical localization of Mira (*Figure 2A*, 2:06) and Mira was driven into the cytoplasm during prophase (*Figure 2A*, 2:21). However, Mira failed to relocalize to a basal crescent following NEB (*Figure 2A*, 2:33). Thus, an actin cytoskeleton is not required for the interphase localization of Miranda, nor its removal from the interphase cortex. However, it is required for Mira basal localization following NEB.

We next tested whether F-actin was required to maintain Mira crescents. To this end, we arrested NBs in metaphase by depolymerising microtubules with colcemid (at which point Mira crescents are established) and then treated them with LatA. In this situation, LatA treatment caused Mira to relocalize to the cytoplasm (*Figure 2A*), indicative of a role for F-actin in retaining Mira basally after NEB.

However, LatA treatment after NEB also led to the redistribution of Baz/Par3 and aPKC to the entire NB periphery. Therefore, the observed effect on Mira could be indirect, caused by changes in aPKC localization when F-actin is compromised. Assuming that aPKC activity is restricted to the cortex (*Atwood et al., 2007*; *Rodriguez et al., 2017*) we sought to distinguish between direct and indirect effects on Mira by determining if Mira loss preceded (indicative of direct effect of loss of actin) or followed (indicative of an indirect effect caused by changes in aPKC localization) changes in Par complex distribution in colcemid arrested NBs upon LatA treatment. We found that Mira loss preceded changes in cortical aPKC/Baz localization in response to LatA. This occurred about 2.8 ± 1

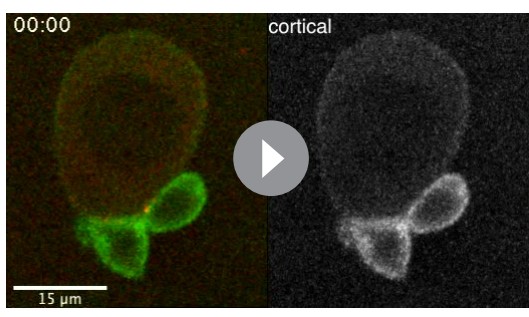

**Video 1.** Interphase cortical Miranda is removed at the onset of mitosis. Spinning disc confocal image of a neuroblast expressing Baz::GFP (red) and Mira:: mCherry (green). For this and all subsequent videos maximum projection after a 3D Gaussian blur (FIJI, radius 8/.8/1) of 7 consecutive equatorial planes taken at 0.4 µm spacing is shown. Z-stacks taken every minute. Time stamp: hh:mm.
DOI: https://doi.org/10.7554/eLife.29939.006

min (n = 13) before aPKC (*Video 4*) or Baz (*Video 5*) became detectable at the basal cortex (*Figure 2—figure supplement 1*). Therefore, changes in cortical aPKC localization upon LatA treatment are unlikely to drive Mira off the cortex. We conclude that an intact actin network facilitates both establishment and maintenance of asymmetric Mira localization in mitotic larval NBs, though it is not clear how actin carries out this function.

One possibility is that actin-associated Myosins (*Barros et al., 2003*; *Erben et al., 2008*; *Petritsch et al., 2003*) stabilize Mira at the basal cortex in mitosis. Therefore, we tested which step of Mira localization involved myosin. Myosin motor activity is enhanced by the phosphorylation of myosin regulatory light chain, encoded by the *sqh* gene in *Drosophila* (*Jordan and Karess, 1997*). We disrupted this phosphorylation using ML-7, a specific inhibitor of myosin light chain kinase (MLCK, *Bain et al., 2003*). As with

LatA, ML-7 treatment of cycling NBs affected neither the uniform interphase cortical localization of Mira (*Figure 2B*, 0') nor its clearance during prophase (*Figure 2B*, 138-278') but resulted in a failure to establish a basal crescent after NEB, which was restored upon drug washout (*Figure 2B*, 328'). In colcemid arrested NBs, ML-7 treatment also resulted in Mira redistributing from the basal crescent to the cytoplasm, which also was restored upon ML-7 washout. However, unlike LatA treatment, ML-7 treatment did not cause the Par complex to spread over to the entire cortex (*Figure 2C*, *Video 6*). Finally, we demonstrated the specificity of the effect of ML-7 by counteracting its effect with the phosphomimetic version of myosin regulatory light chain Sqh$^{EE}$, as in (*Das and Storey, 2014*). Overexpressing Sqh$^{EE}$ significantly delayed loss of cortical Mira after ML-7 addition in colcemid-arrested NBs (*Figure 2D*, *Video 7*).

In summary, these results support the notion that Mira interacts differently with the cortex in interphase and in mitosis since F-actin and myosin activity contribute to establish and maintain asymmetric Mira crescents at the basal cortex following NEB, but they are not essential for uniform cortical localization of Mira in interphase nor for Mira clearance during prophase.

## Miranda binds directly to the plasma membrane in interphase NBs

The observation that Mira continues to localize to the cortex in interphase upon F-actin disruption suggested that it is directly retained at the PM. In *Drosophila* S2 cells Mira binds directly to phospholipids of the PM via its BH motif, and phosphorylation of this motif by aPKC abolishes this binding (*Bailey and Prehoda, 2015*). Therefore, we tested the role of the BH motif in Mira localization in interphase and mitosis.

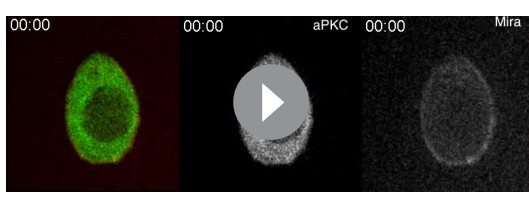

**Video 2.** Interphase cortical Miranda is removed at the onset of mitosis. Spinning disc confocal image of a neuroblast expressing aPKC::GFP (green) and Mira:: mCherry (red). Z-stacks taken every minute. Time stamp: hh:mm.
DOI: https://doi.org/10.7554/eLife.29939.007

Mira's membrane interaction is sensitive to the phosphorylation by aPKC of a Serine which resides in the BH motif (S96), (*Bailey and Prehoda, 2015*). To understand the influence membrane binding has on the dynamic localization in NBs, we used a CrispR based approach to generate four mCherry-tagged *Miranda* alleles (*Figure 3A*): (1) control (S96 unchanged, ctrl, able to rescue embryonic lethality); (2) a phosphomutant (S96A, homozygous embryonic lethal); (3) a phosphomimetic (S96D, homozygous embryonic lethal and shown in vitro to reduce phospholipid binding and Mira recruitment to the PM when overexpressed in S2 cells);

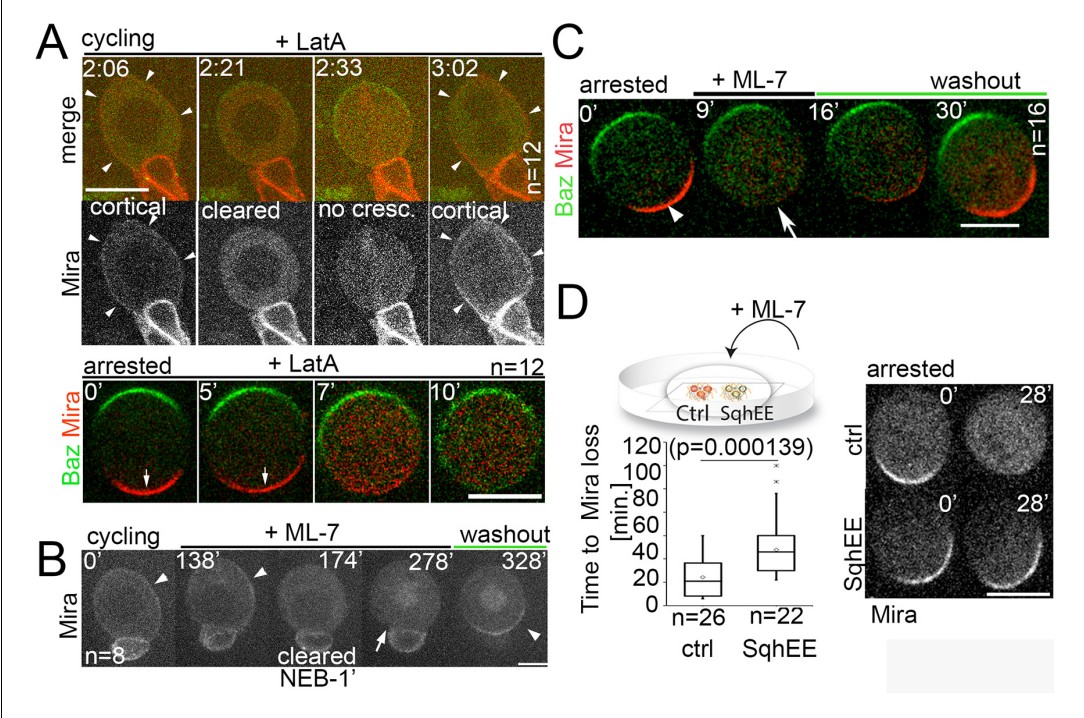

**Figure 2.** Differential response of Mira localization in interphase and mitosis to disruption of the actin cytoskeleton. (**A**) Stills from *Video 3*. LatA was added to a cycling NB in primary cell culture expressing Baz::GFP (green) and Mira::mCherry (red). Arrowheads point at cortical Mira after culturing ~1 hr with LatA (2:06). At 1 min to NEB, Mira::mCherry is cleared from the cortex (2:21). Mira forms no crescent in the next mitosis (2:33), but after cytokinesis fails (note bi-nucleated cell at 3:02), Mira is recruited to the cortex (arrowheads). Bottom panels: Colcemid-arrested NBs expressing Baz::GFP and Mira::mCherry. LatA was added at 5 μM prior to imaging at 15 s. intervals. Mira crescents (arrows) are lost upon LatA treatment. (**B**) Cycling NB in primary cell culture expressing Mira::mCherry, that remains cortical upon ML-7 addition (15 μM; interphase: 0' and 138', arrowheads), is cleared 1 min prior to NEB (174'), does not form a crescent after NEB (278', arrow), but accumulates on the spindle (seen in cross section). After ML-7 washout, a basal Mira::mCherry crescent recovers (arrowhead, 328'). (**C**) Related to *Video 5*. Colcemid-arrested NB in primary cell culture expressing Baz::GFP (green) and Mira::mCherry (red). After addition of 20 μM ML-7 Mira (arrowhead, 0') becomes cytoplasmic (arrow,+9'), but upon ML-7 washout a Mira crescent recovers. (**D**) The effect of 20 μM ML-7 can be quenched by overexpressing a phospho-mimetic form Sqh (Sqh$^{EE}$). Colcemid arrested NBs (ctrl: Mira::mCherry: SqhEE: Mira::mCherry co-expressing SqhEE by worniuGal4). Ctrl and SqhEE were co-cultured and ML-7 added (related to *Video 6*). Quantification of the time required to cause Mira::mCherry to become cytoplasmic shown on the left. Two-tailed t test for independent means revealed significance. *BAC{mira::mcherry-MS2}* was the source of Mira::mCherry. Scale bar: 10 μm.

DOI: https://doi.org/10.7554/eLife.29939.008

The following figure supplement is available for figure 2:

**Figure supplement 1.** Mira falls homogenously off the cortex upon LatA treatment, which is not driven by aPKC cortical displacement.

DOI: https://doi.org/10.7554/eLife.29939.009

and (4) a complete deletion of the BH motif (ΔBH, homozygous embryonic lethal).

The control localized uniformly to the cortex in interphase, was cleared from the cortex in prophase and reappeared as a basal crescent after NEB (*Figure 3A*, *Video 8*). In contrast, the phospho-mutant S96A localized uniformly to the cortex in interphase but was not cleared at the onset of prophase and remained detectable on the entire cortex throughout mitosis, when it was also transiently enriched at the apical pole (possibly because it forms abnormally stable interactions with apically localized aPKC). After NEB, S96A mutant protein remained localized uniformly at the cortex, even in the presence of LatA (*Figure 3A,B*, *Video 9*). In contrast, the phosphomimetic S96D did not localize robustly to the cortex in interphase and instead accumulated predominantly on cortical microtubules, as evidenced by its relocalization to the cytoplasm upon colcemid treatment (*Figure 3A,B*). Nevertheless, S96D always localized asymmetrically, with a basal bias at the cortex following NEB, which appeared to occur at reduced levels compared to controls (*Figure 3A*, *Video 10*).

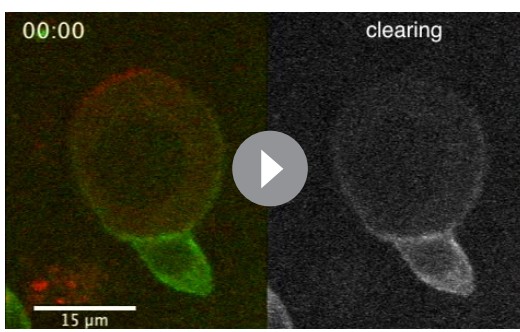

**Video 3.** Interphase cortical Miranda is actin independent. Spinning disc confocal image of a NB expressing Baz::GFP (red) and Mira::mCherry (green) showing a control division before 1 μM LatA was added. Z-stacks taken every minute. Time stamp: hh:mm.

DOI: https://doi.org/10.7554/eLife.29939.010

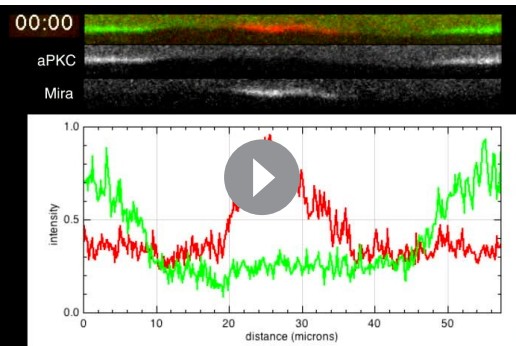

**Video 4.** Colcemid-arrested NB expressing aPKC::GFP and Mira::mCherry that was treated with 5 μM LatA at the beginning of the recording at 16 s intervals. The cortex was straightened out and split at the apical pole such that aPKC::GFP appears right and left and Mira in the center. Fluorescence profiles shown below. Note that Mira falls off homogenously from the cortex and becomes cytoplasmic at 5:36 (red arrowhead), while the detectable borders of cortical aPKC (green arrowheads) have not yet changed. Only from 07:12 onward aPKC rise above cytoplasmic levels where Mira was localized. Time stamp: mm:ss.

DOI: https://doi.org/10.7554/eLife.29939.011

Deletion of the BH motif abolished both uniform cortical localization in interphase and asymmetric cortical localization after NEB, when it was entirely cytoplasmic (*Figure 3A*, *Video 11*, see *Figure 3C* for quantification and *Figure 3D* for summary of the localization of the Mira). Finally, ectopic activation of aPKC by overexpression of constitutively active aPKC$^{\Delta N}$ (*Betschinger et al., 2003*) also prevented Mira cortical localization in interphase and most of mitosis. Of note, Mira localization was rescued in telophase in aPKC$^{\Delta N}$ overexpressing NBs (*Figure 3—figure supplement 1A,A'*), suggesting that even in the presence of deregulated aPKC activity, Mira cortical association is not completely lost.

These findings support the idea that, in interphase, the BH motif is necessary and likely to be sufficient to mediate interactions with phospholipids of the PM leading to uniform cortical localization of Mira. These findings also show that phospho-regulation of the BH motif affects Mira localization in both phases of the cell cycle. Therefore, these observations support the model that Mira uses only one mode, BH motif mediated PM interactions, for cortical association throughout the cell cycle. However, this model does not readily explain differences in the response of Mira to LatA and ML-7 in interphase versus mitosis (*Figure 2*).

### Failure to clear interphase Miranda in *apkc* mutants results in persistence of uniform plasma membrane bound Miranda in mitosis

NBs mutant for *apkc* fail to localize Mira asymmetrically in mitosis (*Rolls et al., 2003*). Since we found that clearance of uniform cortical Mira at the onset of mitosis fails when S96 cannot be phosphorylated, we predicted that in *apkc* mutant NBs at the onset of mitosis, Mira should not be cleared from the PM. To test this, we expressed fluorescently tagged Mira in *apkc*$^{k06403}$ mutant brains; a loss of function condition for aPKC (*Wodarz et al., 2000*). As in controls, Mira localized uniformly to the cortex of interphase *apkc*$^{k06403}$ NBs, but did not clear from the cortex during prophase and remained uniformly localized throughout mitosis (*Video 12*, *Figure 4A*).

From these observations, we made two predictions: First, the abnormal cortical localization of Mira in metaphase in the absence of aPKC should occur independently of F-actin, similar to normal cortical Mira in control interphase (*Figure 2A*). Second, if Mira binding to the cortex is differently controlled in interphase and mitosis, the turnover of Mira at the cortex when bound to the PM in interphase and when interacting with F-actin after NEB should be different. Turnover can be measured by fluorescence recovery after photo-bleaching (FRAP). Mira recovery should be different in interphase and mitosis and controls, but similar between control interphase NBs and mitotic *apkc* mutant NBs, as we suspect Mira to bind the PM in these mutants throughout the cell cycle.

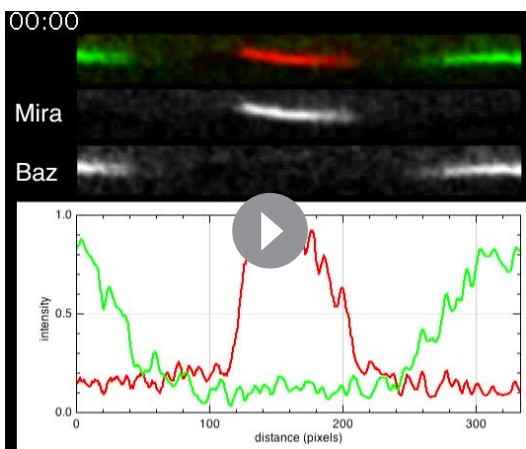

**Video 5.** Colcemid-arrested NB expressing Baz::GFP and Mira::mCherry that were treated with 5 μM LatA at the beginning of the recording at 16 s intervals. The cortex was straightened out and split at the apical pole such that Baz::GFP appears right and left and Mira in the center. Fluorescence profiles shown below. Note that Mira falls off homogenously from the cortex and becomes cytoplasmic at 9:00 (asterisks), while the detectable borders of cortical Baz (arrowheads) have not yet changed. Only from 12:30 onward Baz rise above cytoplasmic levels where Mira was localized. Time stamp: mm:ss.

DOI: https://doi.org/10.7554/eLife.29939.012

Indeed, Mira localization to the cortex in mitosis was insensitive to LatA treatment in NBs depleted for aPKC (*Figure 4—figure supplement 1*). Consistent with the second prediction, FRAP measurements revealed that Mira recovery was significantly different in interphase and mitosis (*Figure 4B–C'*). However, while Mira recovery in mitosis was faster when aPKC was knocked down by RNAi (or aPKC inhibition by Lgl[3A] overexpression, *Betschinger et al., 2003*), this did not result in Mira recovery in mitosis becoming as fast as in interphase (*Figure 4C,C'*). It is known that changes in the actin network caused by progression through the cell cycle (*Ramanathan et al., 2015*) can influence dynamics of membrane-associated proteins in general (*Heinemann et al., 2013*). This is certainly the case in NBs, as a photo-convertible membrane-associated reporter that attaches to the entire NB PM via a myristoylation signal (myr-Eos) showed slower dynamics in mitosis compared to interphase (~four fold, *Figure 4D,D'*). Thus, these general cell cycle-driven changes in the actin cytoskeleton could account for the difference between Mira recovery in interphase and in mitosis in aPKC-impaired NBs.

To test the effect of such changes in the actin cytoskeleton on mobility of PM interacting proteins, we treated mitotic NBs with LatA, which resulted, in myr-Eos dynamics falling within a range similar to that observed for interphase cells (*Figure 4D,D'*). Importantly, myr-Eos dynamics did not change in response to Lgl[3A] overexpression, arguing against Lgl[3A] overexpression causing changes in the actin cytoskeleton to explain the resulting accelerated redistribution of Mira in mitosis. Finally, in mitotic NBs overexpressing Lgl[3A], LatA treatment resulted in Mira recovery becoming as fast as in interphase (*Figure 4C,C'*).

In conclusion, these results show that, in unperturbed NBs, Mira turnover at the PM in interphase and at the basal cortex in mitosis are different, supporting the notion that Mira has different binding modes in interphase versus mitosis. Furthermore, in *apkc* mutant NBs, instead of being cleared, Mira may persist throughout mitosis with the same actin-insensitive uniform localization, and similar turnover, as in interphase.

## High doses of Y-27632 inhibit aPKC and partially disrupt maintenance of Mira asymmetry after NEB

We assessed the relative contributions of actomyosin and aPKC to Mira localization throughout the cell cycle, and show that at the onset of mitosis aPKC displaces Mira from the PM. After NEB, however, Mira localization becomes actomyosin-dependent. We next attempted to address whether aPKC regulates Mira localization after NEB.

To dissect a role for aPKC during the cell cycle, temporal control over its activity is required. This can be achieved with temperature

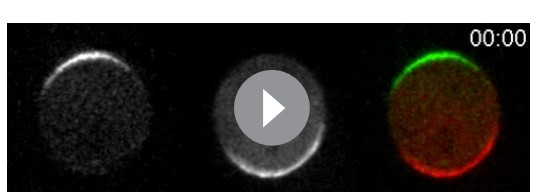

**Video 6.** Myosin inhibition reversibly affects basal Mira anchoring in a polarized NB. A colcemid arrested NB expressing Baz::GFP (green) and Mira::mCherry (red) in primary cell culture was treated with 20 μM ML-7 which was washed out when indicated. Left panel Baz::GFP, middle panel Mira::mCherry, right panel merge. Z-stacks taken every minute. Time stamp: mm:ss.

DOI: https://doi.org/10.7554/eLife.29939.013

sensitive (ts) alleles or small molecule inhibition. We found that the available ts allele of *apkc* (*Guilgur et al., 2012*) is already hypomorphic at permissive temperatures resulting in Mira localization defects (not shown). Therefore, we made use of the non-specific effects of the ROCK inhibitor Y-27632, which inhibits aPKC with an $IC_{50}$ of ~10 μM (*Atwood and Prehoda, 2009*).

We determined the concentration at which Y-27632 treatment phenocopied the *apkc* phenotype, and resulted in LatA insensitive uniform cortical Mira after NEB. This was the case when at least 200 μM Y-27632 was added to cycling NBs (*Video 13*, n = 25, *Video 14*, n = 12). This resulted in only partial loss of Mira asymmetry: Mira signal became detectable faintly at the apical pole 52 ± 11 min (n = 15) following this treatment, but Mira remained basally enriched (*Figure 5A*). This basal enrichment was only lost upon the addition of LatA (*Figure 5A*, *Video 15*, n = 15). Therefore, as reported previously (*Barros et al., 2003*), high doses of Y-27632 lead to Mira localization defects that are likely to reflect aPKC inhibition (*Atwood and Prehoda, 2009*). Intriguingly, there seems to be a difference in sensitivity to Y-27632 when added before or after NEB: Addition of high doses of Y-27632 added to cycling cells results in uniform cortical Mira whereas (*Video 13*), when the same dose is added to metaphase-arrested NBs, Mira appears with a delay (~52 min) and only faintly at the apical pole, but retains a LatA-sensitive basal bias (*Figure 5A*). These results suggest that in addition to the role of aPKC in clearing uniform PM bound Mira at the onset of mitosis, aPKC may contribute to Mira asymmetry after NEB, possibly by clearing it from the apical cortex. However, given the likelihood of several targets for Y-27632, a precise role for aPKC after NEB cannot be determined in this way.

## Low doses of Y-27632 can affect Mira crescent size independently of aPKC inhibition

In the course of determining the lowest concentration of Y-27632 to phenocopy aPKC loss of function, we observed that addition of 50 μM of Y-27632 to metaphase-arrested NBs, did not produce any significant changes in Mira or aPKC crescent size (n = 22, *Figure 5B*, and *Figure 5—figure supplement 1*). However, addition of just 25 μM Y-27632 to cycling NBs, produced aPKC crescent size comparable to controls (*Figure 5—figure supplement 1*) but significantly enlarged Mira crescents (*Figure 5C*).

Furthermore, the enlarged Mira crescents resulting from Y-27632 addition to cycling NBs were sensitive to LatA and ML-7 treatment, as were normal Mira crescents in controls. This suggests that also under this condition, actomyosin is important to retain Mira at the cortex mitosis, even when the size of the crescents is enlarged (*Figure 5D*; see *Figure 5E* for Mira crescent size quantification under the different conditions). Finally, NBs

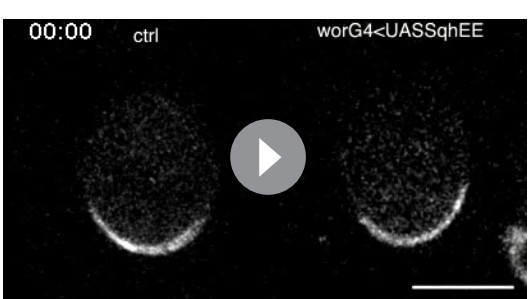

**Video 7.** The effect of ML-7 on cortical Mira localization in mitosis can be delayed by overexpressing Sqh[EE]. Mira::mCherry NB (ctrl) and Mira::mCherry NB co-expressing Sqh[EE] (rescue) were co cultured in neighboring clots in the same dish and the effect of ML-7 on cortical Mira recorded. Z-stacks taken every two minutes. Time stamp: hh:mm.
DOI: https://doi.org/10.7554/eLife.29939.014

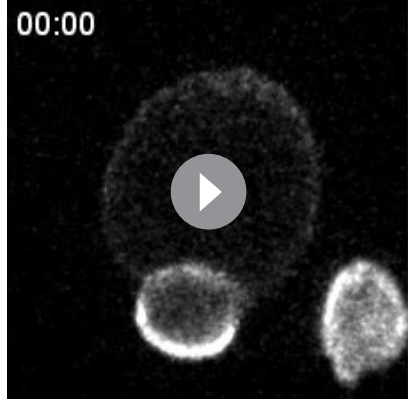

**Video 8.** Control *mira*[mCherry] allele generated by CrispR/Cas9. Mira localizes to the interphase cortex, from where it is cleared before NEB. Then Mira relocalizes to a larger crescent. Therefore this allele and Mira::mCherry (BAC rescue) are undistinguishable in terms of Mira dynamics. This control further shows that the MS2 binding site in the BAC rescue construct does not interfere with Mira cortical dynamics. Time stamp: hh:mm.
DOI: https://doi.org/10.7554/eLife.29939.017

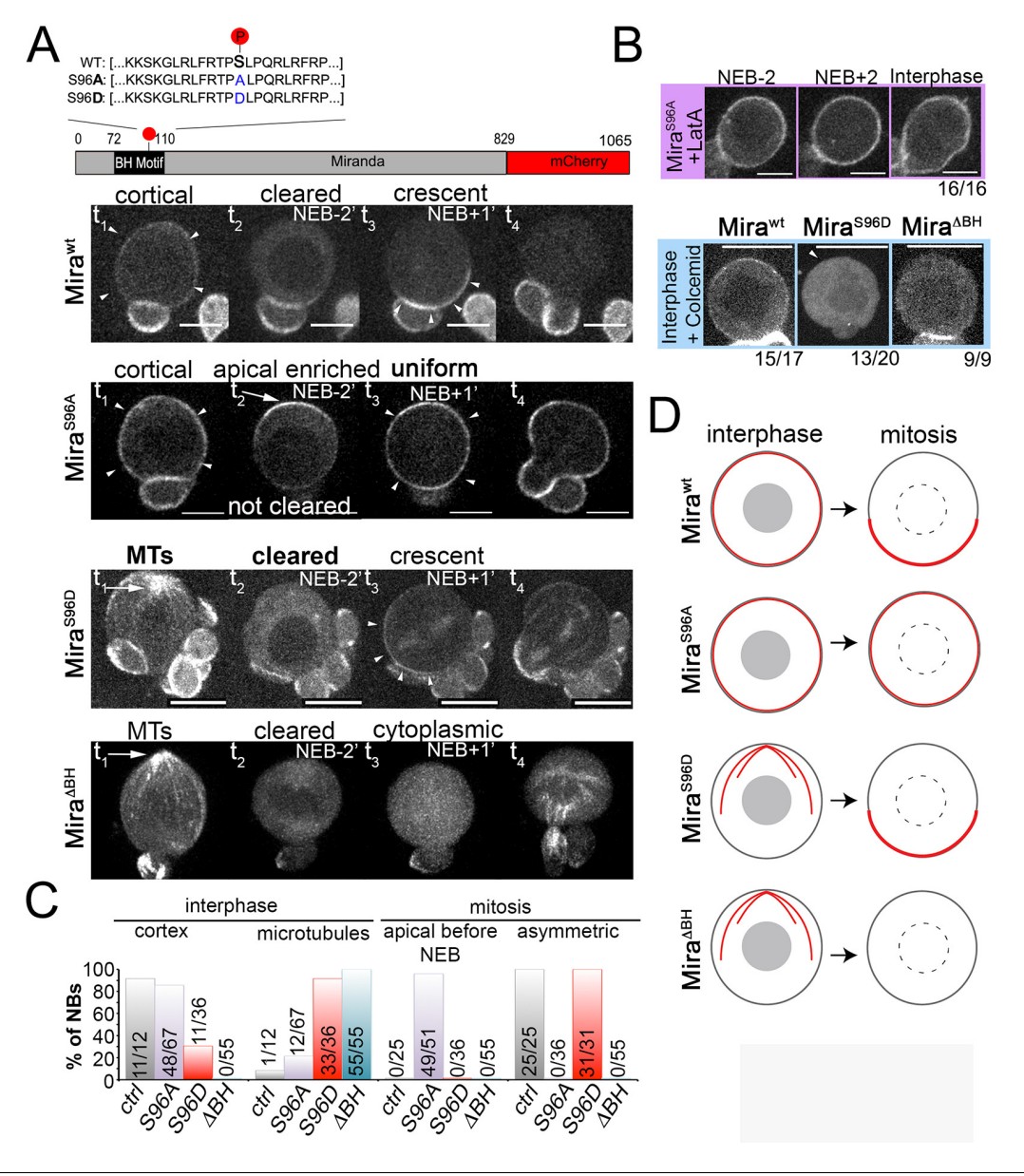

**Figure 3.** Miranda binds to the plasma membrane in interphase NBs via its BH motif. (**A**) Schematic indicating the different Mira alleles used. Mira::mCherry localizes cortically uniform in interphase (arrowheads $t_1$), is cleared from the cortex shortly before NEB and forms a crescent (arrowheads $t_3$) thereafter that is inherited by daughter cells (related to *Video 8*). The phosphomutant S96A is uniformly cortical in interphase, accumulates apically shortly before NEB (arrow, $t_2$), and is uniformly cortical after NEB (arrowheads $t_3$) and in telophase ($t_4$, related to *Video 9*). The phosphomimetic S96D localizes to cortical microtubules in interphase (arrow $t_1$), is cleared from the cortex before NEB and asymmetric after NEB (arrowheads $t_3$) and segregates to daughter cells (related to *Video 10*). Deletion of the BH motif leads to cortical microtubule localization in interphase (arrow $t_1$), cytoplasmic localization before and after NEB and reappearance on microtubules around cytokinesis (related to *Video 11*). (**B**) Neuroblasts expressing the indicated Mira alleles were treated with 1 μM LatA or 50 μM colcemid for 60 min. Cortical localization of S96A is insensitive to LatA treatment. Below: While the control remains cortical, S96D and ΔBH become cytoplasmic upon colcemid treatment. (**C**) Frequency of indicated localization of the different Mira mutants. (**D**) Schematic of the localization of the different Mira alleles. Scale bar: 10 μm.

DOI: https://doi.org/10.7554/eLife.29939.015

The following figure supplement is available for figure 3:

**Figure supplement 1.** Effects of aPKC$^{ΔN}$ expression in NBs on Mira localization.

*Figure 3 continued on next page*

*Figure 3 continued*

DOI: https://doi.org/10.7554/eLife.29939.016

that divided in the presence of 25 µM Y-27632 produced larger daughter cells than controls (n = 12, *Figure 5C,F*). Therefore, enlarged Mira crescents induced by Y-27632 are correlated with an increase of NB daughter cell size.

In conclusion, while Y-27632 at higher concentration, can indeed mimic the effect of *apkc* mutation on Mira, these results suggest that when NBs polarize in the presence of low concentrations of Y-27632, Mira crescent size is affected, which is likely to occur independently of aPKC inhibition. These results suggest that Mira cortical retention has different mechanisms of regulation in interphase and in mitosis. They also hint at an additional, Y-27632 sensitive layer of regulation controlling basal Mira crescent size.

## Discussion

In this study, we addressed the localization of the adapter protein Mira throughout the cell cycle of *Drosophila* larval NBs to shed light on how polarized fate determinant localization is achieved. It was previously demonstrated that actomyosin is essential for Mira polarization in mitosis (*Barros et al., 2003*; *Petritsch et al., 2003*; *Shen et al., 1998*). More recent work showed that Mira can directly bind to the PM via its BH motif. As a result, spatially controlled phosphorylation of this BH motif by aPKC, leading to displacement of Mira from the cortex where aPKC is active, can in principle explain Mira asymmetry without the need to evoke a role for actomyosin (*Atwood and Prehoda, 2009*; *Bailey and Prehoda, 2015*).

To address this apparent inconsistency, we reassessed in vivo the relative contribution of aPKC and actomyosin throughout the cell cycle analysing Mira localization using endogenously expressed reporters in living NBs. This has allowed us to resolve this problem as we find that asymmetric Mira localization is established stepwise and involves both aPKC-dependent phosphorylation and actomyosin-dependent anchoring, which are required at different time points in mitosis.

We propose that Mira has two different modes by which it can be retained at the cortex (*Figure 6*). In interphase, Mira localizes uniformly to the cortex via direct interactions with the PM for which its BH motif is necessary and likely to be sufficient and which occurs independently of an intact F-actin cortex (*Figure 2A*). After NEB, Mira still relies on the BH motif to localize in a basal crescent, but at this stage of the cell cycle it might be required to mediate actomyosin-dependent basal retention of Mira (*Figure 3A*, *Figure 2A–D*). The transition between these localizations depends on phosphorylation by aPKC (*Figure 3A*, *Figure 4A*).

We observe that deletion of the BH motif as well as overexpression of aPKC$^{\Delta N}$ disrupt cortical localization of Mira in interphase and mitosis (*Figure 3A* and *Figure 3—figure supplement 1*) and that the phosphomimetic S96D mutation reduces Mira localization in interphase as well as in mitosis (*Figure 3A*). These findings by themselves argue for the model that throughout the cell cycle Mira cortical association depends solely on BH motif-mediated interaction with the PM, that is negatively regulated by locally controlled aPKC phosphorylation (*Atwood and Prehoda, 2009*; *Bailey and Prehoda, 2015*).

What could be the role of F-Actin for Mira localization in this model? F-Actin clearly contributes to aPKC regulation of Miranda localization by restricting the localization of the Par complex to the apical pole as LatA addition changes the distribution of aPKC and Baz (*Figure 2A*). However, at least with the resolution with which we can observe live NBs, basal Miranda relocates into the cytoplasm in LatA treated metaphase arrested NBs before changes in the localization of the Par complex are induced (*Figure 2—figure supplement 1*). Furthermore, inhibiting Myosin activity in metaphase arrested NBs with ML-7 also caused the relocation of Mira into the cytoplasm. However, the localization of the Par-complex was unchanged (*Figure 2B,C*). This argues that actomyosin plays an additional anchoring function that contributes to retain Mira basally after NEB.

The intensity (*Figure 1B,B'*) and turnover (*Figure 4C,C'*) of Mira differ in interphase and mitosis. If BH motif mediated retention at the PM sufficed to mediate Mira cortical binding throughout the cell cycle, these observations might be explained by changes in Mira properties such as dimerization. Mira can form homo-dimers (*Yousef et al., 2008*) and targeting of proteins with phospholipid-

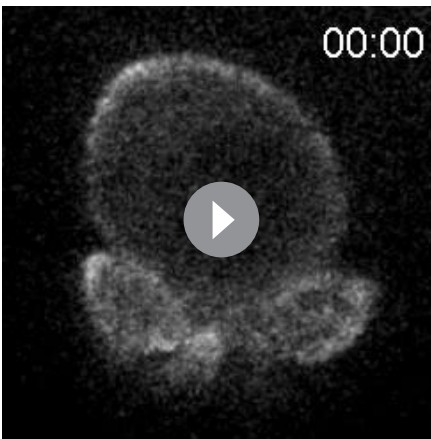

**Video 9.** Phosphomutant S96A allele of Mira tagged with mCherry at the C-terminus. Mira localizes uniformly to the interphase cortex. Shortly before NEB, S96A is apically enriched, before being uniformly cortical after NEB and during division. Time stamp: hh:mm.
DOI: https://doi.org/10.7554/eLife.29939.018

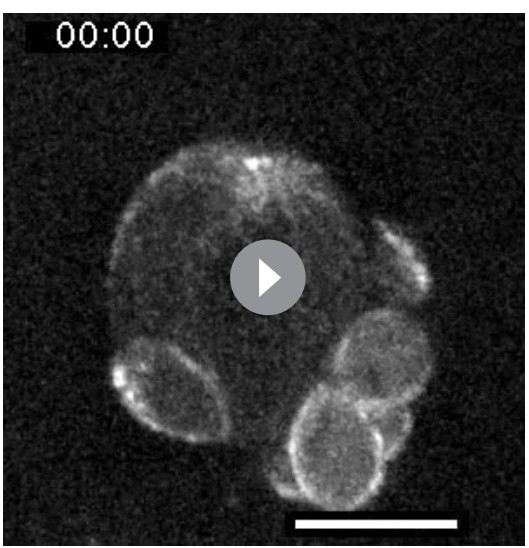

**Video 10.** Phosphomimetic S96D allele of Mira tagged with mCherry at the C-terminus. S96D localizes to microtubules in interphase, but is asymmetric in mitosis. Note the signal resembling subcortical microtubules in interphase converging at the apical pole. After NEB a basal crescent is detectable. At 115:30 a z-stack spanning the entire NB was collected and the maximum projection is frozen. After this, 50 µM colcemid was added to reveal if Mira$^{S96D}$::mCherry binds to the cortex. Next frozen frame: similar stack after 30 min in colcemid. Next frozen frame: 50 min in colcemid – no cortical signal is detectable. Last frozen frame 65 min in colcemid. Time stamp: mm:ss. Scale 15 µm.
DOI: https://doi.org/10.7554/eLife.29939.019

binding domains to membranes often requires their dimerization (*Lemmon, 2008*). However, a point mutation preventing dimerization of Mira's cargo binding domain (required to bind Pros) does not prevent Mira's asymmetric localization in mitosis, while Pros localization is lost (*Jia et al., 2015*). We found that Pros localizes to the PM in a Mira-dependent manner in interphase (*Figure 1—figure supplement 1*), suggesting that Mira is already a dimer at this stage. Thus, changes in dimerization are unlikely to explain the differences in Mira turnover detectable by FRAP.

Differences in turnover may be explained by different modes of cortical association of Mira in mitosis and interphase. We propose that in mitosis, additional stabilizing interactions might retain Mira basally, which are not operating in interphase. Those stabilizing interactions require the actomyosin network, but rather than being pushed towards the basal pole by Myosin (*Barros et al., 2003*), actomyosin may provide a Mira anchoring function (*Figure 6*). Mira might directly bind to F-actin as shown in vitro (*Sousa-Nunes et al., 2009*) or be anchored basally by myosin activity. Alternatively, additional processes could be involved such as Mira's interaction with *mira* mRNA (*Ramat et al., 2017*), potentially providing an anchoring scaffold to maintain Mira basally. The phosphomimetic S96D mutation strongly disrupts uniform localization to the PM in interphase, but localizes at the basal cortex in mitosis, albeit at reduced levels (*Figure 3A*). This is consistent with the existence of Mira stabilizing interactions in mitosis, that are not present in interphase, which reduce the effect of a negative charge provided by the Aspartate in the phosphomimetic mutant on cortical Mira localization.

A surprising finding is that the BH motif is essential for Mira localization in interphase and in mitosis. In mitosis, BH motif mediated PM binding is no longer sufficient to localize Miranda to the basal pole. This is indicated by the requirement of the actomyosin cytoskeleton after NEB (*Figure 2C,D*). However, the BH motif is still necessary for Mira localization after NEB. It is possible that the BH-phospholipid interactions still play a role in mitosis. Deletion of the BH motif could also cause more indirect effects. For example, MiraΔBH is not found on mitotic microtubules, where Mira is typically observed in conditions where it is unable to localize correctly (*Albertson and Doe, 2003*;

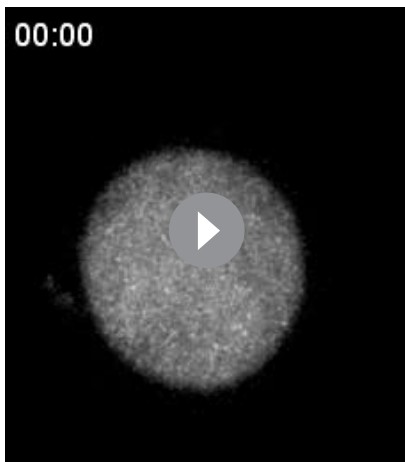

**Video 11.** Mira requires its BH motif for interphase cortical localization (see main text) and basal localization in mitosis. The BH motif in Mira has been deleted by gene editing and this Mira mutant tagged with mCherry at the C-terminus (*mira*<sup>ΔBHmCherry</sup>). Mira<sup>ΔBH</sup>::mCherry when homozygous is found on the interphase microtubule network and in the cytoplasm during mitosis. Time stamp: hh:mm.
DOI: https://doi.org/10.7554/eLife.29939.020

Barros et al., 2003; Rolls et al., 2003; Slack et al., 2007). We propose that the BH motif may mediate Mira's interaction with acto-myosin, which remains to be tested.

Interfering with aPKC-dependent Mira displacement from the cortex during prophase, either by *apkc* mutation or by directly preventing phosphorylation of the BH motif (S96A) results in the persistence of uniform, PM bound Mira in metaphase; indicated by its abnormally fast dynamics and its actomyosin independence (*Figure 4C,C'* and *Figure 4—figure supplement 1*). This aPKC-dependent step in prophase might be a prerequisite for basal crescent formation in metaphase. One possibility is that phosphorylation of the BH motif might potentiate Mira's ability to engage with actomyosin for basal retention after NEB. Mira phosphorylation might need to be properly balanced and locally controlled to allow for Mira asymmetric localization. This could explain why the phosphomimetic S96D mutant, displays reduced basal localization in metaphase and that upon overexpression of aPKC<sup>ΔN</sup> Mira does not form basal crescents in metaphase (*Figure 3—figure supplement 1*). Another unexpected observation is that despite never localizing to the PM in interphase or to the cortex in metaphase, Mira localization is rescued at telophase in aPKC<sup>ΔN</sup> overexpressing NBs, which does not occur when the BH motif is deleted (*Video 11*). This suggests that Mira retains at least in telophase the ability to engage with the cortex when aPKC<sup>ΔN</sup> is expressed and that the BH motif might be important for the telophase rescue phenomenon (*Peng et al., 2000*).

In addition to its role in displacing Mira into the cytoplasm during prophase, aPKC might contribute to Mira localization after NEB. High doses of Y-27632 when added to colcemid arrested NBs lead to apical and lateral accumulation of Mira at the cortex (*Figure 5A*), suggesting that also after NEB aPKC contributes to keep Mira off the apical membrane. However, at 200 μM Y-27632 is likely to inhibit multiple processes. Therefore, the precise contribution of aPKC at different time points during the cell cycle remains to be determined.

Compartmentalized aPKC activity provides an explanation for Mira crescent size in a model in which, throughout the cell cycle, Mira solely relies on BH mediated PM interactions to localize to regions of the cortex where aPKC is inactive. This would hold true in a model where Mira uses one mode to interact with PM in interphase and another to bind to actomyosin in mitosis, if phosphorylation of the BH motif was required for basal Mira retention by actomyosin after NEB. An interesting possibility is that spatial information for Mira crescent size is provided by the actomyosin network itself. Low doses of Y-27632 yield enlarged basal Mira crescents that are correlated with an increase in daughter cell size (*Figure 5C, E,F*), the control of which involves actomyosin regulation (*Roubinet and Cabernard, 2014*).

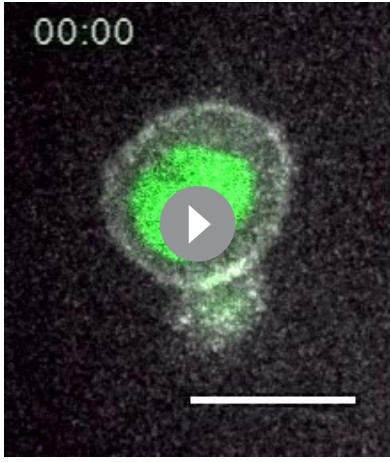

**Video 12.** Miranda remains at the cortex throughout the cell cycle in *apkc*<sup>k04603</sup> mutant NBs. Mutant NB, labeled with nlsGFP (green) expressing Mira::mCherry (white). Z-stacks taken every minute. Time stamp: hh:mm.
DOI: https://doi.org/10.7554/eLife.29939.023

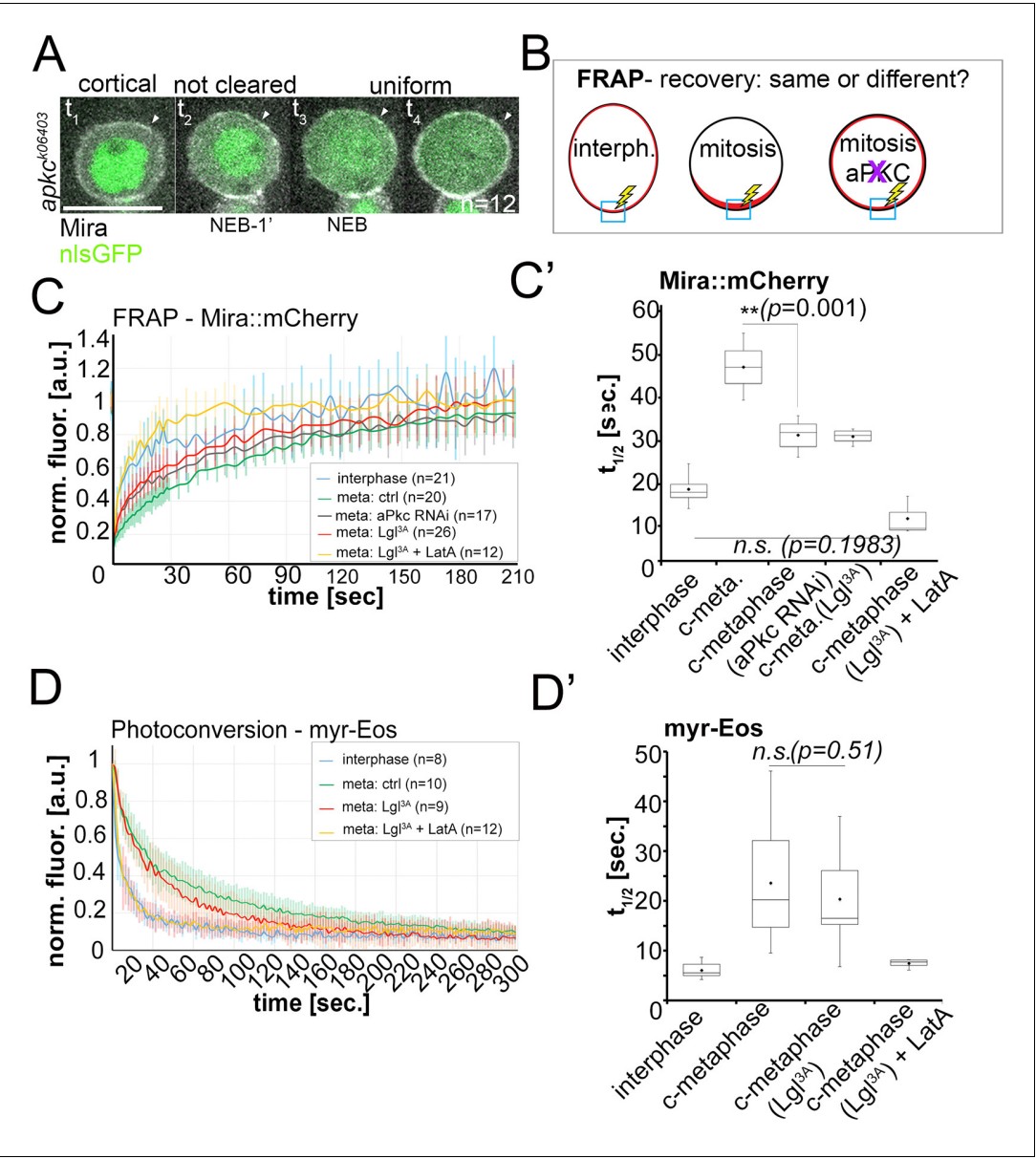

**Figure 4.** Lateral diffusion and cytoplasmic exchange of cortical Miranda are different in control and aPKC impaired mitotic NBs. (**A**) Stills from **Video 12** of an *apkc^k06403* mutant NB (MARCM clone labeled with nlsGFP, green) expressing Mira::mCherry (grey). Mira is cortical in interphase, as the NB enters mitosis, and after NEB (arrowheads, $t_1 - t_4$). (**B**) Conditions analyzed by FRAP. (**C**) Fluorescence redistribution curves of cortical Mira:: mCherry at the indicated conditions. (**C'**) Estimates of $t_{1/2}$ [sec.] for cortical Mira::mCherry under the indicated conditions derived from curve fitting (**Rapsomaniki et al., 2012**). (**D**) Photo-conversion experiment monitoring loss of myr-EOS converted signal over time. (**D'**) Estimates of $t_{1/2}$ [sec.] for cortical Mira::mCherry under the indicated conditions from curve fitting. Overexpression was driven by worniu-Gal4. p values: two-tailed t test for independent means. Scale bar: 10 µm.

DOI: https://doi.org/10.7554/eLife.29939.021

The following figure supplement is available for figure 4:

**Figure supplement 1.** Mira localization to the mitotic NB cortex occurs independently of F-actin upon aPKC knock down.

DOI: https://doi.org/10.7554/eLife.29939.022

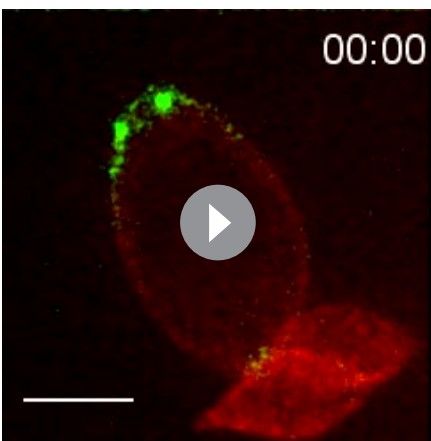

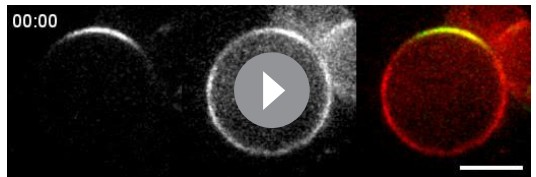

**Video 13.** Miranda remains at the entire cell cortex throughout the cell cycle in NBs treated with 200 µM Y-27632. A Baz::GFP (Green) and Miranda::mCherry (Red) NB was imaged through one cell cycle in the presence of 200 µM Y-27632. Miranda remained cortical throughout while Bazooka still localised apically in mitosis (n = 12). Z stacks taken every 2 min. Time stamp: hh:mm. Scale: 10 µM.
DOI: https://doi.org/10.7554/eLife.29939.026

**Video 14.** 200 µM Y-27632 induced uniform cortical Mira in mitosis localizes independently of an intact actin network. A Baz::GFP and Mira::mCherry expressing NB was cultured in the presence of 200 µM Y-27632 and then arrested with colcemid. 5 µM LatA was added after the first frame of the movie. LatA induces loss of Baz asymmetry, yet Mira remains cortical. Z-stacks shown. Z stacks taken every 2 min. Time stamp: hh:mm.
DOI: https://doi.org/10.7554/eLife.29939.027

Therefore, spatial information for determinant localization in NBs could be coupled to the machinery that regulates daughter cell size. Indeed, ROCK has recently been shown to accumulate at the apical pole in prophase generating an asymmetry in the actomyosin network (*Tsankova et al., 2017*).

The $IC_{50}$ of Y-27632 for ROCK is about an order of magnitude lower than the $IC_{50}$ determined for aPKC (*Atwood and Prehoda, 2009*; *Uehata et al., 1997*). Therefore, enlarged Mira crescents we observe for 25 µM Y-27632 on Mira in cycling NBs might stem from effects on ROCK. These may trigger changes in actomyosin configuration and/or cortical tensions (*Matthews et al., 2006*; *Tsankova et al., 2017*). It is thus possible that local tension anisotropies in the actomyosin network provide spatial information for Mira localization. ROCK and MLCK both affect myosin activity (*Amano et al., 1996*; *Saitoh et al., 1987*; *Ueda et al., 2002*; *Watanabe et al., 2007*) yet the effects of MLCK (ML-7) and ROCK (Y-27632) inhibition on Mira differ (*Figure 2* versus *Figure 5*). In MCDK II cells, Y-27632 and ML-7 treatment had different effects on myosin regulatory light chain phosphorylation (*Watanabe et al., 2007*). This could result in different effects on myosin activity that may explain the different effects on Mira also in NBs.

What could be the advantage of relying on PM binding in interphase and actomyosin retention in mitosis? Nuclear levels of Mira's cargo Pros in NBs affect quiescence and differentiation (*Choksi et al., 2006*; *Lai and Doe, 2014*). It is possible that PM-bound Mira sequesters Pros at the interphase NB PM. Two modes of cortical retention might allow regulation of nuclear Pros levels in interphase NBs and ensure segregation of elevated determinant levels to daughter cells in mitosis to achieve correct thresholds of cell fate information in the differentiating daughter cells.

## Materials and methods

### Fly stocks and genetics

Flies were reared on standard cornmeal food at 25°C. Lines used were:

(1) Baz::GFP trap (*Buszczak et al., 2007*); (2) $w^{1118}$ (Bloomington); (3) MARCM: hsFlp tubGal4 UASnlsGFP; FRT42B tubGal80/Cyo and FRT82B gal80 (*Lee and Luo, 1999*); (4) worniu-Gal4 (*Albertson et al., 2004*); (5) UAS-Lgl$^{3A}$ (*Betschinger et al., 2003*); (6) w1118, y,w, hsp70-flp; tubP-FRT >cd2>FRT-Gal4, UAS-GFP (Gift from M. Gho); (7) Mz1061 (*Ito et al., 1995*); (8) UAS-GFP::Mira (*Mollinari et al., 2002*). (9) FRT82B mira$^{L44}$ (*Matsuzaki et al., 1998*). (10) Df(3R)ora$^{I9}$ (*Shen et al., 1997*). (11) UAS-aPKC$^{RNAi}$: *P{y[+t7.7] v[+t1.8]=TRiP.HMS01320}attP2* (BL#34332); *(12)* Numb::GFP (*Couturier et al., 2013*); (13) FRT42B apkc$^{k06403}$ (*Wodarz et al., 2000*); (14) UAS-aPKC$^{ΔN}$; (15) P {UASp-sqh.E20E21}3 (BL#64411); (16) P{10xUAS-IVS-myr::tdEos]attP2 (BL #32226); y[1] w[*]; P{y[+t*] w[+mC]=UAS-Lifeact-Ruby}VIE-19A (BL# 35545); (17) aPKC::GFP (*Besson et al., 2015*); *Source 1 of*

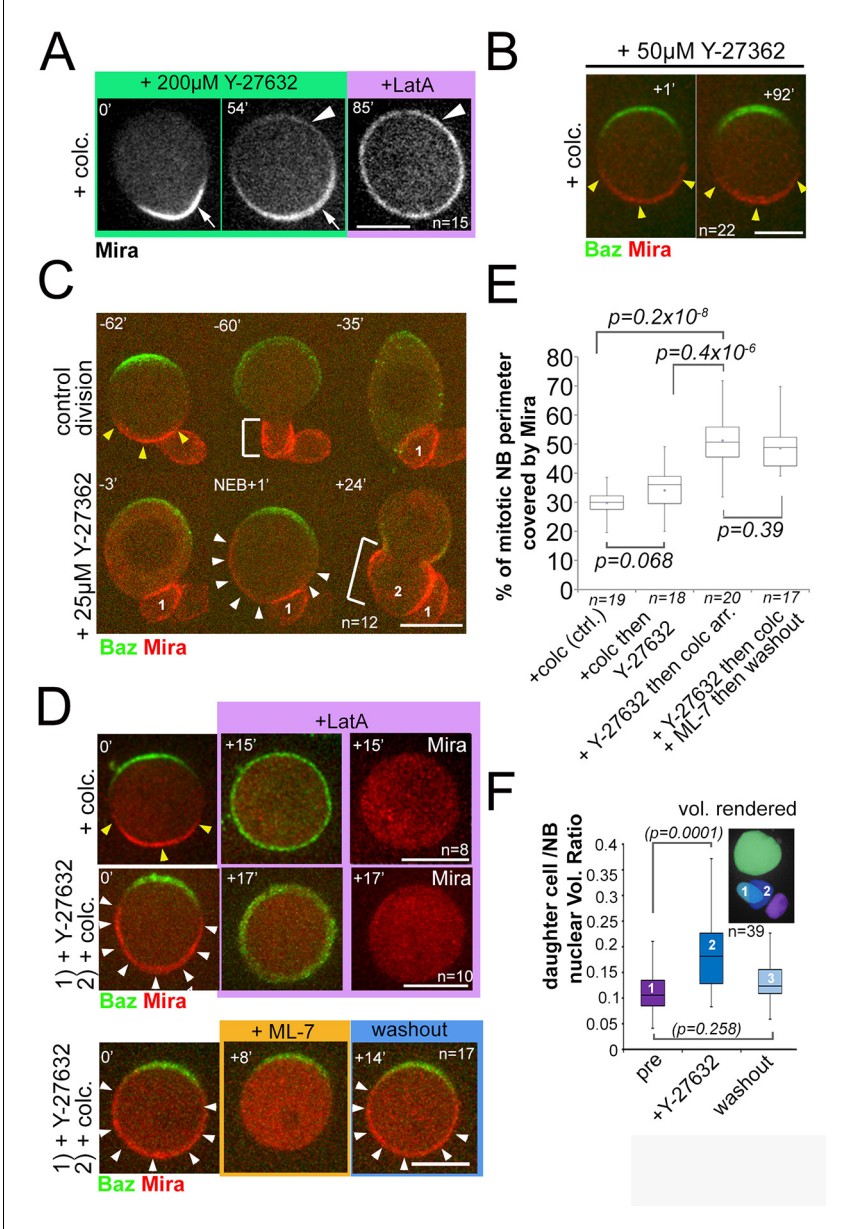

**Figure 5.** Mira crescent size is affected by a Y-27632-sensitive mechanism that operates before NEB. (A) Stills from *Video 15*. Colcemid arrested NBs were treated with 200 µM Y-27632. After >50 min Mira becomes faintly detectable apically, but retains a basal bias. LatA addition (5 µM) abolishes that asymmetric bias and Mira is uniformly distributed on the membrane. (B) Culturing colcemid-arrested NBs in 50 µM Y-27632 did not alter Mira crescent size (yellow arrowheads, quantified in E). (C) NBs polarizing in the presence of 25 µM Y-27632 show enlarged Mira crescents. Control division (−62′ to −35′) with normally sized Mira crescent and daughter cell size (−60′; yellow arrowheads, bracket, respectively). Dividing in the presence of Y-27632 (−3, NEB +1) leads to an enlarged Mira crescent (NEB +1, white arrowheads) and enlarged daughter cell size (+24′, brackets, 2). (D) NBs were allowed to polarize in the absence (*upper row*) or presence of 25 µM Y-27632 (middle and lower row) followed by colcemid arrest. *upper row*: Control NB with normal Mira crescent (yellow arrowheads) was depolarized by 1 µM LatA. Mira was displaced into the cytoplasm. *middle row*: adding 1 µM LatA leads to displacement of the enlarged Mira crescent (yellow arrowheads) in the cytoplasm. *Lower row:* adding 20 µM ML-7 drives Mira into the cytoplasm (+8′). Upon ML-7 washout, Mira recovered to an enlarged crescent (+14′, white arrowheads). (E) Quantification of Mira crescent size in the aforementioned experiments (unpaired t test). (F) Plot of the ratio of daughter cell to NB nuclei as a measure of the effect of Y-27632 on daughter cell size. NBs expressing NLSGFP were imaged by DIC to follow daughter cell birth order during three consecutive divisions [(1) pre-treatment; (2) division in the presence of 25 µM Y-27632; (3) division after drug washout]. A high-resolution

*Figure 5 continued on next page*

*Figure 5 continued*

z-stack of nlsGFP was recorded, and the nuclear volumes rendered and calculated using IMARIS to plot their ratio. *p* values: Dunn's test. *Time* stamp: min. Labels as indicated. *BAC{mira::mcherry-MS2}* was the source of Mira:: mCherry. Scale bar: 10 μm.

DOI: https://doi.org/10.7554/eLife.29939.024

The following figure supplement is available for figure 5:

**Figure supplement 1.** Standard used to quantify Mira crescent size.

DOI: https://doi.org/10.7554/eLife.29939.025

---

*Mira::mCherry: BAC{mira::mcherry-MS2}* (**Ramat et al., 2017**). aPKC[RNAi] clones were generated by heat shocking larvae of the genotype *y,w, hsp70-flp; tub-FRT >cd2>FRT-Gal4, UAS-GFP; P{y[+t7.7] v[+t1.8]=TRiP.HMS01320}attP2*. Heat shocks were performed 24hph and 48hph for 1 hr at 37°C. MARCM clones were generated by heat shocking L1 larvae for 2 hr at 37°C.

Generation of Mira alleles: Source 2 of Mira::mCherry: *mira*[mCherry]; *mira*[ΔBHmCherry] (**Ramat et al., 2017**), *mira*[S96AmCherry] and *mira*[S96DmCherry] are derived from *mira*[KO] (**Ramat et al., 2017**). *mira*[mCherry] was generated by inserting a modified wt genomic locus in which mCherry was fused to the C-terminus following a GSAGS linker into *mira*[KO]. For *mira*[S96DmCherry]: TCG (Serine96) was changed to GAC (aspartic acid). For *mira*[S96AmCherry]: TCG was replaced with GCG (alanine). CH322-11-P04 was the source for the *mira* sequences cloned using Gibson assembly into the RIV white vector (**Baena-Lopez et al., 2013**) that was injected using the attP site in *mira*[KO] as landing site. *BAC{mira:: mcherry-MS2}* (**Ramat et al., 2017**) see **Figure 1—figure supplement 2**). While *mira*[mCherry] behaves similarly to *BAC{mira::mcherry-MS2}* and rescues embryonic lethality, *mira*[ΔBHmCherry], *mira*[S96AmCherry] and *mira*[S96DmCherry] are homozygous lethal.

Live imaging: Live imaging was performed as described (**Pampalona et al., 2015**). Briefly, brains were dissected in collagenase buffer and incubated in collagenase for 20 min. Brains were transferred to a drop of fibrinogen (0.2mgml⁻1, Sigma f-3879) dissolved in Schneider's medium (SLS-04-351Q) on a 25 mm Glass bottom dish (WPI). Brains were manually dissociated with needles before the fibrinogen was clotted by addition of thrombin (100Uml$^{-1}$, Sigma T7513). Schneider's medium supplemented with FCS, Fly serum and insulin was then added. A 3–4 μm slice at the center of the neuroblasts was then imaged every 30–90 s using a 100x OIL objective NA1.45 on a spinning disk confocal microscope. Data were processed (3D Gaussian blur 0.8/0.8/0.8 pixels) and analyzed using FIJI (**Schindelin et al., 2012**). Nuclear volume was measured using Imaris software. All other drugs were added to the media either prior to or during imaging: ML-7 (Sigma, I2764, dissolved in water), Y-27632 (Abcam, Ab120129, dissolved in water). Drugs were washed out by media replacement; in the polarity reconstitution assay colcemid concentrations were kept constant throughout the experiments. FRAP experiments were carried out on a Leica SP8 confocal using a 63x NA1.2 APO water immersion objective. To estimate $t_{1/2}$ for the recovery curves, we used published curve fitting methods (**Rapsomaniki et al., 2012**).

## Immunohistochemistry

### Primary larval brain neuroblast cell culture

Brains were dissected in collagenase buffer and incubated for 20 min in collagenase, as for live imaging. Brains were then transferred into supplemented Schneider's medium and manually dissociated by pipetting. Cells were pipetted onto a poly-lysine-coated 25 mm glass-bottomed dish and left to adhere for 40 min. Schneider's was then replaced with 4% formaldehyde (Sigma) in PBS and cells were fixed for 10 min. Cells were permeabilized with 0.1% PBS–Triton for 10 min. Cells were then washed with PBS 3 × 10 min before antibody staining overnight at 4°C. All antibodies were dissolved in PBS–1%Tween.

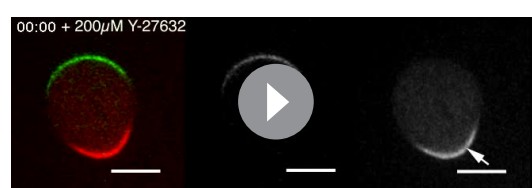

**Video 15.** Colcemid arrested NB expressing Baz::GFP and Mira::mCherry, treated with 200 μM Y-27632. Mira starts to become visible ~36 min after Y-27632 addition in this example, but remains asymmetrically distributed, until LatA is added. Z-stacks shown. Z stacks taken every 2 min. Time stamp: mm:ss.

DOI: https://doi.org/10.7554/eLife.29939.028

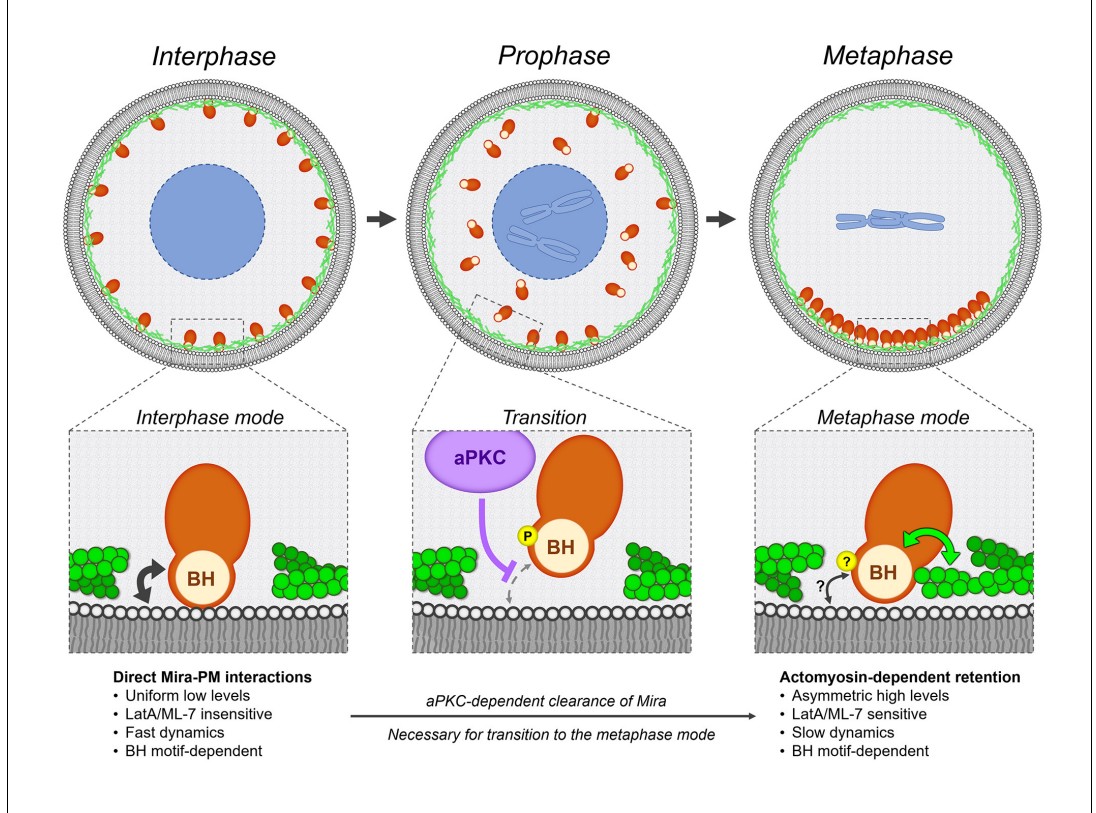

**Figure 6.** Model. Mira associates with the cortex using two different modes, which characteristics are detailed in the bottom row. During interphase, Mira directly binds to the phospholipids of the PM *via* its BH motif (black double arrow). During prophase, aPKC-dependent phosphorylation of this motif abolishes this interaction, resulting in the progressive clearance of Mira from the cortex, in an apical-to-basal manner driven Mira into the cytoplasm. This clearance in prophase is necessary for Mira to associate with the basal cortex after NEB, *via* Actomyosin-dependent retention. Both the precise phosphoregulation and molecular characteristics of this mode remain to be determined. The BH motif, also required at this step, may directly or indirectly mediate interactions between Mira and actomyosin (green double arrow). PM interactions *via* its BH motif (black double arrow) may still contribute, but are not sufficient to mediate Mira basal retention after NEB.
DOI: https://doi.org/10.7554/eLife.29939.029

*Whole mount brains*: Brains were fixed in 4% formaldehyde (Sigma) for 20 min at room temperature. Primary antibodies: Rabbit anti-Miranda (1:200, gift from C. Gonzalez); Mouse anti-GFP (1:400, Abcam); Rabbit anti-Brat (1:200, a gift from J. Knoblich); Guinea pig anti-Dpn (1:500 a gift from J. Skeath); Mouse anti-Pros (1:40, DSHB). To stain F-actin we used Alexa Fluor 488 or 561 coupled Phalloidin (Molecular Probes, 5:200) for 20 min at room temperature. Secondary antibodies were from Life Technologies and raised in donkey: Anti-rabbit Alexa-594; Anti-mouse Alexa-488; Anti-rabbit Alexa-647; Anti-guinea pig Alexa-647. Microscopy was performed using a Leica-SP8 CLSM (60x Water objective, 1.2) and images were processed using FIJI.

In all cases the sample size *n* provided reflects all samples collected for one experimental condition. All experimental were repeated at least twice.

## Acknowledgements

We thank C Doe, J Knoblich, F Schweisguth, D StJohnston, F Matsuzaki, C Gonzalez, J Skeath, A Wodarz, M Gho and the Kyoto and Bloomington stock centers for reagents and/or protocols. We thank A Müller, C Weijer, M Gonzalez-Gaitan, T Tanaka and K Storey for critical reading. MRH is supported by an MRC PhD studentship. We thank the Dundee imaging facility for excellent support. Work in JJ's laboratory is supported by Wellcome and the Royal Society Sir Henry Dale fellowship 100031Z/12/Z. MRH is supported by an MRC studentship funded by these grants: G1000386/1, MR/

J50046X/1, MR/K500896/1, MR/K501384/1. The tissue imaging facility is supported by the grant WT101468 from Wellcome.

## Additional information

### Funding

| Funder | Grant reference number | Author |
|---|---|---|
| Medical Research Council | G1000386/1 | Matthew Robert Hannaford |
| Medical Research Council | MR/J50046X/1 | Matthew Robert Hannaford |
| Medical Research Council | MR/K500896/1 | Matthew Robert Hannaford |
| Medical Research Council | MR/K501384/1 | Matthew Robert Hannaford |
| Wellcome Trust | 100031Z/12/Z | Anne Ramat<br>Nicolas Loyer<br>Jens Januschke |
| Royal Society | 100031Z/12/Z | Jens Januschke |

The funders had no role in study design, data collection and interpretation, or the decision to submit the work for publication.

### Author contributions

Matthew Robert Hannaford, Conceptualization, Data curation, Formal analysis, Writing—original draft; Anne Ramat, Data curation; Nicolas Loyer, Data curation, Formal analysis, Writing—original draft; Jens Januschke, Conceptualization, Data curation, Formal analysis, Supervision, Funding acquisition, Writing—original draft, Project administration, Writing—review and editing

### Author ORCIDs

Matthew Robert Hannaford http://orcid.org/0000-0001-7772-0450
Anne Ramat http://orcid.org/0000-0002-2103-9583
Nicolas Loyer http://orcid.org/0000-0001-5010-2564
Jens Januschke http://orcid.org/0000-0001-8985-2717

### Decision letter and Author response

Decision letter https://doi.org/10.7554/eLife.29939.031
Author response https://doi.org/10.7554/eLife.29939.032

## Additional files

### Supplementary files
• Transparent reporting form
DOI: https://doi.org/10.7554/eLife.29939.030

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
