## [Decision Letter]

[Editors’ note: a previous version of this study was rejected after peer review, but the authors submitted for reconsideration. The first decision letter after peer review is shown below.]

Thank you for submitting your work entitled "Switching the cortical binding mode is required for asymmetric fate determinant localization in *Drosophila* neuroblasts" for consideration by *eLife*. Your article has been reviewed by three peer reviewers, one of whom, Yukiko M Yamashita (Reviewer #1), is a member of our Board of Reviewing Editors, and the evaluation has been overseen by a Senior Editor. The following individual involved in review of your submission has agreed to reveal their identity: Shigeki Yoshiura (Reviewer #2).

Our decision has been reached after consultation between the reviewers. Based on these discussions and the individual reviews below, we regret to inform you that your work will not be considered further for publication in *eLife*.

Whereas all reviewers agreed that the idea of Miranda having two distinct modes of localization is interesting and of potential impact, they also felt that your data do not prove your hypothesis: whereas your data indicate that the existing model of Miranda localization may require a revision, your alternative hypothesis does not provide a better explanation than the current model, leaving unexplained observations as much as the current model does. The individual comments can be found at the bottom of this letter.

The reviewers felt that it is critically important for you to carefully assess the current model and all the data that support it. At least the revised model must be better than the current one in explaining all the observations/data in the field. For example, a key result supporting the current model is that the unphosphorylatable Miranda (Mira5A) is depolarized in mitosis. This would be inconsistent with your revised model. In reaching a new model, it is not sufficient to point out that the current model is imperfect, but it is important that there is a significant advancement in the understanding, instead of muddying the field.

Reviewer #1:

The major conclusion of this paper is that Miranda, a fate determinant that is segregated to the differentiating daughter during asymmetric neuroblast division in *Drosophila*, has two binding modes to the plasma membrane. First, during interphase, Mira localizes to the cortex (evenly). In G2/M, it localizes to the basal cortex in an actin dependent manner. As opposed to the previous model, in which aPKC was suggested to promote basal cortex localization by apical exclusion of Mira through phosphorylation, the present study shows that aPKC is required to clear Mira from cortex prior to its localization to the basal cortex. (i.e. Mira first disappear from the entire cortex, and reappear as a basal crescent in mitosis)

This work revises the current model of Miranda localization in NBs, and shows that Miranda cortical localization is regulated by two distinct mechanisms (even, cortical localization in interphase and basal crescent in mitosis). The authors' main message seems to be that these two mechanisms are independent of each other. The experimental data presented in this manuscript seem to leave rooms for alternative interpretations. Although this paper is right in that revision of the old model is required, this paper does not necessarily present 'better' revised model, as it still contains a few loose ends. Some of the inconsistencies I found in this paper are the following:

1) If Mira localization is regulated by two independent mechanisms, the significance of interphase cortical localization becomes unclear. The reason why Mira localization is important is because it functions as a fate determinant for NB daughter (GMC), thus only mitotic localization should matter.

2) However, the Figure 4 data (Mira-deltaBH cannot go to mitotic basal cortex, whereas Mira-SD goes to basal crescent, despite not being able to localize to the interphase cortex) seems to suggest an interdependence between interphase and mitotic localization. For example, Mira has to localize to the cortex in interphase so that it can be phosphorylated by aPKC, then it dissociates from the cortex but potentiated for basal cortex localization.

Overall, the data is of high quality, and clearly suggest the need of revision to the current model, but the study does not put forward an alternative, revised model that is more convincing than the old one. I do agree that the revision is needed to the current model, but all the data do not fit yet into a simpler model (i.e. something is still missing). The alternative model that they suggest to replace the existing model still leaves unexplained observations. As I am not exactly in this field, I cannot suggest what would be the way to go. I feel the work is important, but this work needs a bit more to be able to say that they have 'better' understanding of the process.

Reviewer #2:

This manuscript by Dr. Januschke and colleagues examined the mechanisms underlying the asymmetric localization of Miranda in the *Drosophila* larval neuroblasts. They proposed that there would be two different mechanisms that regulate the Miranda cortical localization; direct binding to the plasma membrane in interphase and actin-myosin dependent mechanism in mitotic phase. The concept that Miranda utilizes two different mechanisms to localize at the cell cortex in different cell cycle phases is new and interesting, however, there are some points to be clarified before published.

Major comments

1) In the most of the figures, authors utilized Lgl3A overexpression to abolish the function of aPKC, however, as shown in Figure 2, the majority of the Lgl3A expressing neuroblasts achieved asymmetric localization of Miranda in mitosis while in the aPKC mutant clones Miranda localized at the entire cortex (Figure 4), suggesting that Lgl3A overexpression does not recapitulate aPKC loss of function. Although Lgl3A is thought to suppress aPKC function, it seems that, at least in these cases, Lgl3A expression is not enough to suppress aPKC function or it has some other functions than aPKC suppression, making it difficult to interpret Lgl3A overexpression phenotypes. Thus, it is crucial to check what is observed in Lgl3A overexpression is also the case in aPKC mutant or RNAi clones.

2) In Figure 4, the authors showed that Mira[S96D] showed uniform cortical localization in colcemid and LatA treated neuroblast. Following the authors' model, Miranda localizes at the cortex by either direct binding to the plasma membrane or actin-dependent mechanism, however, in this situation Mira seems to localize at cell cortex in the absence of both machinery. Then how does Miranda localize at cell cortex in this situation? Authors should discuss about this point.

3) Authors utilized Y-27632 to temporally inhibit aPKC function but it is not clear whether Y-27632 treatment is really affecting the aPKC activity or not. As a control experiment, it is better to examine whether the addition of 50uM Y-27632 to the cycling neuroblasts results in the Lat-A-insensitive entire cortical localization of Miranda in mitotic phase (like aPKC loss of function situation).

Reviewer #3:

This revision, like the original, proposes that Miranda is polarized in *Drosophila* neuroblasts via two distinct mechanisms, one in interphase and one in mitosis – the former being by a previously described "BH" association mechanism, and the latter being a "myosin affinity zone". I had two main criticisms of this proposed mechanism. First, the authors did not separate these two functions, rather deletion of the BH caused loss of Miranda polarity in both interphase and mitosis. Second, the proposal of a "myosin affinity zone" is based on data using inhibitors leaving open the possibility that the effect is indirect. The author's work is potentially exciting because there are interesting questions related to Miranda's function in interphase, and the role of myosin in polarizing Miranda. Unfortunately, however, I do not believe that the revised manuscript comes close to adequately addressing the criticisms raised in the original review and therefore I do not recommend publication.

• A separation of function allele would be the most definitive test of the authors model – if two different mechanisms are used, then it should be possible to find Miranda alleles that only polarize in mitosis and not in interphase (unless the interphase mechanism is required for the mitotic one, which is actually what the title says – see below). Mira∆BH isn't polarized in interphase or mitosis indicating that the authors model is wrong or the BH motif is required for each binding mode. To distinguish between these possibilities, the authors examined MiraS96D, a phosphomimetic allele, and conclude that it is a bona fide separation of function allele, disrupting interphase localization but not mitotic. However, S96D is clearly a hypomorph (Figure 2 Current Biology 19, 723-729, 2009) and my interpretation of the author's data (i.e. Video 9) is consistent with it behaving as a hypomorph in their assay – Mira96D basal crescent signal is reduced and spindle signal is increased not only in interphase but also in mitosis.

• I also criticized the original version of the manuscript because it proposed that mitotic Miranda polarization occurs through a "myosin anchor" in a "basal affinity zone" because the data supporting this relies on drug treatments. Nothing in the revised manuscript addresses this criticism. The author's rebuttal letter states, "We agree that the mode of action of myosins remains perhaps obscure since we do not know whether the effect is direct or indirect." That is precisely my criticism, that the mode of action of myosins remains obscure. The letter goes on to state, "However, given that the ML-7 effect on Mira basal localization in colcemid arrested NBs can be tempered by overexpressing a phospho-mimetic version of Sqh…, an involvement of myosin activity in anchoring Mira at the basal pole at NEB seems very likely." This statement is not very impactful if the involvement of myosin activity is very indirect, which the authors themselves acknowledge is entirely possible. The letter further states, "We hope that the reviewer would agree that the current apical exclusion model does not leave much room to explain the role of myosin activity in Mira localization…". I strongly disagree with this statement. First, pretty much any model would allow for an indirect role for the cytoskeleton. But more importantly, the authors do not seem to understand the "current apical exclusion model", at least as it is articulated in Bailey and Prehoda, Dev Cell, 2015, which they have referenced. In this model, BH motifs cooperate with accessory interactions to mediate cortical recruitment of Par substrates (e.g. see Discussion section, "Multivalent Interactions Mediate Par Substrate Cortical Localization"), and that aPKC phosphorylation of the BH motif disrupts this interaction. In this model an "accessory interaction" could certainly require interactions with myosin – the key point being that any such interaction must not be sufficient for cortical targeting (otherwise BH phosphorylation wouldn't have an effect on cortical localization). So, if the authors identified a direct interaction between Miranda and myosin that was required for cortical targeting, that would in an of itself not be inconsistent with the "current apical exclusion model", although it would certainly be interesting. All of this leads me to believe that the authors are attempting to frame the impact of their work in the context of refuting a current model of polarity which, in fact, they do not understand.

• The title of the paper is incorrect, according to the authors. The title states, "Switching the cortical binding mode is required for asymmetric fate determinant localization in *Drosophila* neuroblasts". However, the authors claim to have a Miranda allele that doesn't switch cortical binding modes and is still asymmetrically localized, directly refuting the statement made in the title. Stating one thing in the title and another in the text will confuse readers and make it very difficult for those in and outside the field to understand the impact of the paper. Furthermore, it suggests that the authors themselves do not understand the impact of their work.

• Along those lines, like the original manuscript, the revised version is difficult to follow, especially for those outside the field. For example, the current Abstract states, "Here we test the current model of how the Par-complex component aPKC directs the localization of Miranda". This places a large burden on readers, especially those outside the field, to have a familiarity with current polarity models. In my opinion, it is ultimately up to the authors to decide if they want to make their work accessible, but the journal may feel differently.

[Editors’ note: a second version of this study was rejected after peer review, but the authors submitted for reconsideration. The decision letter after peer review is shown below.]

Thank you for submitting your work entitled "aPKC-mediated displacement and actomyosin-mediated retention polarize Miranda in *Drosophila* neuroblasts" for consideration by *eLife*. Your article has been reviewed by four peer reviewers, one of whom, Yukiko M Yamashita (Reviewer #1), is a member of our Board of Reviewing Editors, and the evaluation has been overseen by a Senior Editor. The following individuals involved in review of your submission have agreed to reveal their identity: Shigeki Yoshiura (Reviewer #3).

Our decision has been reached after consultation between the reviewers. Based on these discussions and the individual reviews below, we regret to inform you that your work will not be considered further for publication in *eLife*.

As you remember, in the previous round of your submission, all reviewers agreed that your work has a good potential but poor presentation/writing, over-interpretation and some questionable interpretation precludes publication at that point. We hoped that you take our advice and revise the manuscript drastically. In this round, we have invited the new reviewer per your request, who provided positive comments, yet also felt that the manuscript requires considerable revision to be acceptable. Later in the discussion among reviewers, this reviewer commented " I agree that the poor presentation is major problem, and that they do not convincingly prove the existence of a basal affinity zone.", favoring "giving the authors the chance to resubmit a radically revised version."

We all remain agreed that your discovery that Miranda localization has two modes is interesting and important. However, poor presentation of manuscript (writing makes it unclear what is the most important discovery of this manuscript. Manuscript contains over-interpretations) makes it difficult to accept without 'radical revision'.

As you know, *eLife* does not allow revision unless it is straightforward. This does not exclude your future opportunity to submit the revised manuscript to *eLife* again. However, we sincerely hope that you take our advice to improve the manuscript.

Reviewer #1:

This is a revised version of previously revised manuscript by Januschke and colleagues. They report the molecular mechanism of asymmetric Miranda localization in *Drosophila* neuroblast.

Same as the previous versions of this manuscript, I found that each experiment is well done and well documented. However, the writing and presentation again makes it difficult for me to assess general impact of this manuscript (thus I have to rely on other reviewers with more expertise in the field). The main difficulty in reading this manuscript for me was 1) sometimes it is not clear whether all of their conclusions are consistent among themselves, 2) sometimes it is not clear whether their claims are indeed consistent with previous findings especially when they argue against existing models. (examples will be provided below). Here, I am not trying to be nit-picky about writing, but the lack of clarity throughout the text has made it difficult for me (and probably for other reviewers in the previous rounds) to grasp the (potential) impact of this paper. With that being said, I am not saying that writing only can be a reason to accept or reject a paper.

In addition, in the course of revisions, the authors often responded to reviewer comments only in their rebuttal/response letters, without incorporating it into the main text. This has also made it difficult to relate how the concern was resolved in a way presentable to the future readers. Response letters should be in the format of 1. Reviewer comments, 2. response by authors, 3. Explanation how the authors incorporated reviewer comments and responses within the main text ("based on these[…]we changed the main text as following.[…]"). In the previous versions, all the reviewers understood that the manuscript clearly demonstrates that the current model needs to be revised. However, the reviewers also commented that the authors' new model may not be consistent with the existing data. In response, the authors provided explanations only in the response letter, without changing much in the main text. The authors should keep in mind that the future readers may have the same questions as reviewers (very likely), and thus such questions from reviewers are better be addressed within the main text, not in the rebuttal letter.

Reviewer #2:

This manuscript contains a series of well-controlled and careful experiments that convincingly demonstrate that Miranda localises by different mechanisms in interphase and mitosis. The observations that Miranda is excluded from the cortex by aPKC and retained in a basal crescent by the acto-myosin cortex are not novel, but most previous work has presented these as competing models to explain how Miranda is asymmetrically localised. Thus, the main novelty in this manuscript is that both are true, but at different time points during the cell cycle. While I agree with the other referee's comments, I don't think that any of them challenge this basic conclusion, which in my opinion is sufficiently important to merit publication in *eLife*.

While I am very positive about these results and the quality of the data, I think that the presentation needs to be improved. My version was entirely lacking Figure 3 (which contains one of the key pieces of data showing that Miranda behaves differently in mitosis), and the Videos were not labelled in the same way for downloading as in the text, which made it difficult to relate the two. The text also needs to be improved to highlight the main conclusions and discuss how these relate to previous work. To give just one example, the manuscript presents strong evidence that myosin activity is required for the basal recruitment of Miranda after NEBD (ML-7 treatment, rescue of this effect with SqhEE, low doses of Y-27632), but does not discuss these data in light of previous work from Barros et al. arguing that myosin excludes Miranda from the cortex or the argument against Barros et al. that high concentrations of Y-27632 inhibit aPKC. Januschke et al. seem to have cleared up all of this confusion by using a different drug, proving that it is specific for myosin activation and by separating the effects of Y-27632 on Rho kinase and aPKC at different doses, but they fail to put this all together into a single argument in the discussion that makes their contribution clear.

Reviewer #3:

This manuscript by Januschke's group proposed that the asymmetric Miranda localization in the asymmetric cell division of the *Drosophila* neuroblast is mediated by two mechanisms: cortical exclusion through the phospho-regulation by aPKC and cortical retention by actomyosin network. The concept that Mira utilizes two different mechanisms in different mitotic phases is very interesting but their data does not seem to fully support their idea and seem to leave some room for alternative interpretations.

First of all, there was no Figure 3 in this manuscript.

I think we cannot accept such incomplete manuscript for a peer-reviewing process, but I at least leave some comments.

In the Lgl3A over-expression experiments, as authors mentioned in the letter, the degree of the aPKC suppression seems to be varied; 7/25 seem to completely suppress aPKC and 4/25 seem to fail to suppress aPKC function. I think it is natural to treat the rest 14/25 as the intermediate between these two; a partial suppression of aPKC function. In such situation, " Mira initially asymmetrically localized, but LatA treatment cause Mira to become uniformly localized on the PM". This result is mostly same as the result when authors introduced MiraS96D in which " MiraS96D::mCherry always achieves asymmetric localization in mitosis…MiraS96D::mCherry relocalizes to the entire cortex upon LatA treatment".

Thus, it is highly possible that MiraS96D is a hypomorphic allele with regards to the phospho-regulation by aPKC.

In this point of view, I think these results would not support the authors' idea that " the BH motif mediates two different binding modes", and rather it is possible to interpret that Mira in mitosis would require both PM binding and other supportive interactions mediated by actomyosin for basal localization.

In the most of the figures, authors used Baz-GFP as a marker for the Par complex localization. Baz localization and aPKC localization are not always the same as shown in Figure 6. Thus, it would be important to examine the distribution of aPKC itself. Especially, it is important to examine the kinetics of aPKC in the presence of LatA (Figure 2), because this is the only one result that distinguishing whether Mira was "cleared" by aPKC or "anchored" by actin.

It is difficult to understand the different outcomes of the ML-7 treatment and Y-27362 treatment. Myosin inhibition resulted in the displacement of Mira from the cortex and Myosin/actin inhibition resulted in the expansion of Mira crescent, why? Authors should mention their interpretation about these results.

Reviewer #4:

This manuscript proposes a mechanism for the polarization of the fate determinant Miranda in *Drosophila* neuroblasts. The current model for this process is that Miranda directly interacts with the plasma membrane and phosphorylation of the membrane-interacting peptide by aPKC displaces Miranda into the cytoplasm. The model proposed in the work under consideration is that Miranda polarization occurs via different mechanisms in interphase and mitosis, and the mitosic mechanism involves a "basal affinity zone" that somehow involves the cytoskeleton. In general, I feel that the authors have overinterpreted their data, which relies on drug treatments and in some cases, used flawed logic to draw their conclusions. Some of the main problems are summarized below:

• While the characteristics of interphase localization are interesting, they are not significant unless interphase localization is somehow important for Miranda function

• Drug treatments, such as LatA, lead to loss of Miranda cortical localization, which could result from drug-induced ectopic aPKC activity. The experiment the authors provide to discount this possibility ("anchoring vs. clearing") is not convincing for a number of reasons, including that Baz localization is used as a proxy for aPKC localization. The cortical localization of Miranda in LatA treated neuroblasts lacking aPKC function are consistent with a misinterpretation in this regard.

• The authors over interpret the S96D mutant, which only partially abrogates plasma membrane binding. They state in the text that the localization of mutant is the same as WT but the Videos they provided do not support this claim.

• Even if the previous problems are overlooked, the authors have not gone far enough to characterize the new interactions that they are proposing take place. The phenotype they observe when inactivating the cytoskeleton could arise from very indirect effects, yet they are proposing direct interactions with a "basal affinity zone". I do not believe that they have presented data supporting their claim that the zone is only basal, or that tell us anything about the interactions that would occur in the zone.

• The text is poorly-written and convoluted. It would be very difficult to follow for someone outside the field.

In summary, I do not believe that the authors have adequately disproven the possibility that the Miranda phenotypes they observe in drug-treated neuroblasts arise from indirect effects, especially through effects on the Par complex. Thus, they have not sufficiently demonstrated the existence of a "basal affinity zone". Furthermore, they have not sufficiently characterized the interactions that they propose are occurring in the "basal affinity zone", if it does indeed exist.

[Editors’ note: what now follows is the decision letter after the authors submitted for further consideration.]

Thank you for submitting your article "aPKC-mediated displacement and actomyosin-mediated retention polarize Miranda in *Drosophila* neuroblasts" for consideration by *eLife*. Your article has been reviewed by three peer reviewers, one of whom, Yukiko M Yamashita (Reviewer #1), is a member of our Board of Reviewing Editors, and the evaluation has been overseen by K VijayRaghavan as the Senior Editor. The following individual involved in review of your submission has agreed to reveal their identity: Daniel St Johnston (Reviewer #2).

The reviewers have discussed the reviews with one another and the Reviewing Editor has drafted this decision to help you prepare a revised submission.

As you can see in the individual review comments, they agreed with the potential importance of your discovery, yet even the most positive reviewer shows a concern about the interpretation of the results. In addition, during reviewer discussion, all agreed that the presentation (writing, flow of the logic) requires considerable improvement such that clear and consistent message can be delivered to the readers. We must emphasize that the reviewers had major concerns regarding the fact that the presentation has not improved in the past rounds of submission-review cycles. Please pay serious attention in these aspects in preparing your revision.

Reviewer #1:

The authors describe interesting behavior of Miranda localization during *Drosophila* neuroblast divisions: Miranda directly binds to plasma membrane (and evenly around the neuroblast cortex) during interphase, it is cleared from the plasma membrane at transition to mitosis, and comes back as 'basal crescent' in mitosis, which is required for asymmetric outcome of the neuroblast division.

They show that:

-In interphase, Mira directly bins to plasma membrane (evenly) in an actin independent manner.

-F-actin/myosin-dependent localization of Mira operates only in mitosis.

-Defective clearance of interphase cortical Miranda by aPKC leads to a failure in asymmetric Miranda localization in mitosis.

This is a revised, new submission of previously rejected manuscript at *eLife*, and this revised version has some additional data that solidify their claims. Their data may provide interesting insights into Mira localization throughout NB cell cycle, clarifying certain aspects of existing models on Mira localization.

However, the major weakness of this manuscript in writing remains. Aside from multiple clear grammatical mistakes, the lack of contexts when they describe their data really interferes readers' understanding (the most important question in the field that is addressed by each specific experiment is not explained). I listed up several specific places, where the way it's written confused me a lot. I recommend the authors to go through extensive editing focusing on 'reader friendly writing'. Given the history of this submission, we must insist that the next version will be the last. Please be certain that you have had others read and proof this so that the Board and reviewers will have an easier time deciding if you have made your essential points. Failing that, the paper will not be returned for another rounds of revisions.

Reviewer #2:

This manuscript contains a series of well-controlled and careful experiments that present evidence that Miranda localises by different mechanisms in interphase and mitosis. The observations that Miranda is excluded from the cortex by aPKC and retained in a basal crescent by the acto-myosin cortex are not novel, but most previous work has presented these as competing models to explain how Miranda is asymmetrically localised. Thus, the main novelty in this manuscript is that both may be true, but at different time points during the cell cycle. The experiments are carefully quantified and informative, but more care is needed in the interpretations of the results. The authors should discuss alternative explanations for their data, even if they are not their preferred model. Another obvious weakness is the use of Y-27632 as an aPKC inhibitor when it also affects myosin activity through ROCK, although this experiment is still valuable and provides a useful comparison with previous work. The manuscript is an improvement over the previous version, but needs to be revised to include a more comprehensive discussion of the meaning of the results that considers alternative interpretations and that is more accessible to a general audience.

Specific comments:

1) FRAP stands for Fluorescence Recovery after Photobleaching and not Fluorescence Redistribution.

2) Intriguingly, such enlarged crescents were LatA and ML-7 sensitive, both indicators that enlarged Mira crescents were not due to lack of phosphorylation of Mira by aPKC. This needs more explanation and perhaps some discussion as to whether the expansion depends on aPKC phosphorylation of another substrate.

Reviewer #3:

This revision from Hannaford et al. remains seriously flawed and I do not recommend it for publication. First, many of the author's conclusions are contradicted by their own data. For example, a key conclusion is that Miranda is cleared from the cortex at prophase. It's difficult or impossible to make this assessment from the data the authors show because of Miranda signal from attached GMCs. Video S2 (included for a different reason) shows a beautiful division without any visible GMCs. This video supports a conclusion opposite of the author's: Miranda remains on the cortex throughout the cell cycle including in prophase. Another example: if Miranda is cleared all at once from the cortex upon LatA treatment, as the authors conclude, then why does the kymograph show the Miranda crescent shrinking? This effect is obfuscated by compressing the transition over a small number of pixels but Video S4 shows it beautifully. Scrubbing video S4 back and forth around the time that Miranda disappears clearly shows the crescent growing and shrinking, not disappearing all at once. Another example: The MLCK inhibitor ML-7 causes cortical Miranda to be lost in metaphase, but the phenotype could be non-specific (SqhEE expression only causes delay of the phenotype), and even if it isn't this one result doesn't prove that Miranda loss is due to a direct effect. Thus, close inspection of the author's data does not support clearing of Miranda at prophase or loss of anchoring by cytoskeletal poisons (in fact, for two experiments the opposite conclusion is supported). The paper contains a fair amount of data, such as the difference in photobleaching dynamics of Miranda in interphase and mitosis, and the examination of Y-27632 treated neuroblasts, which appear to be superfluous without a more clear explanation.

Considering that the paper's foundational conclusions are flawed, it is not surprising that the strongest tests of the resulting model (that Miranda is cleared from the cortex by aPKC phosphorylation of its BH motif during prophase and anchored by actomyosin during metaphase) fail miserably. These tests come in three experiments. First, Miranda∆BH – if the BH is only required for interphase localization, then Miranda∆BH should be cytoplasmic and interphase and polarized during metaphase. However, the authors find that it's cytoplasmic in metaphase too. Expression of constitutively active aPKC (aPKC ∆N) is also a great test of the model. If aPKC activity is just required to clear interphase cortical localization, constitutively active aPKC should result in cytoplasmic Miranda in interphase and polarized Miranda in metaphase. The result again contradicts the model's prediction: Miranda is cytoplasmic in both. Remarkably the authors don't even discuss the metaphase result in the paper! Finally, Miranda S96D (reduces but does not ablate BH interactions with the membrane) should have reduced cortical interactions in interphase but polarized normally in metaphase. Here again the result, reduced cortical signal in both interphase and metaphase, is not consistent with the author's proposed model.

Remarkably, these results directly contradict the author's proposed model but are precisely as predicted from a simpler model in which aPKC regulated BH interactions with the membrane control Miranda cortical association throughout the cell cycle. Of course the more complex model could be saved by making it even more complex – perhaps the BH is also involved in the actomyosin interaction, perhaps aPKC also regulates the actomyosin anchor, perhaps a single point mutant that partially disrupts the BH lipid interaction also partially disrupts the actomyosin interaction – and the authors make a half-hearted attempt to do so (at least for the first experiment), but more data would need to be provided to support a more complex model (e.g. how aPKC might regulate the anchor).

I believe the problems outlined above represent deep and fundamental flaws and therefore preclude publication in its current form. I also note that the authors removed the model figure from the paper. In my opinion this is moving in the wrong direction and encourage the authors to revise the paper so it is more clear, not less.

[Editors' note: further revisions were requested prior to acceptance, as described below.]

Thank you for submitting your article "aPKC-mediated displacement and actomyosin-mediated retention polarize Miranda in *Drosophila* neuroblasts" for consideration by *eLife*. Your article has been reviewed by two peer reviewers, and the evaluation has been overseen by a Reviewing Editor and K VijayRaghavan as the Senior Editor. The reviewers have opted to remain anonymous.

The reviewers have discussed the reviews with one another and the Reviewing Editor has drafted this decision to help you prepare a revised submission.

The core discovery of this manuscript 'Miranda localization is regulated differently during interphase and mitosis' is interesting and worth publishing. Knowing that Miranda localization switches the mode (plasma membrane-bound in interphase, and anchored via actin cytoskeleton in mitosis) is an important stepping stone to fully understand how Mira localization is regulated to achieve asymmetric neuroblast division.

The reviewers had a lengthy discussion as to how to proceed with this manuscript. First, we agreed that the core discovery described above is worth publishing if accurately stated: that Miranda localization have two modes in interphase and mitosis. However, while several reviewers agreed that the data in the manuscript sufficiently supported their model, we were not able to reach a consensus on this point. In the end, the reviewers agreed to recommend publication provided that the authors clearly and accurately state their results in a manner that allows critical assessment of all the data, such that all the readers can judge on their own.

All reviewers noted concerns regarding the accuracy of descriptions. More specific comments will follow below, but the collective result of inaccurate statements/descriptions is that 1) the real contribution of the manuscript is blurred, 2) the manuscript reads as if the authors are proposing an alternative model which they claim to be better than the existing model, but the new model also leaves many unexplained observations, making readers wonder whether the new model is a real improvement or only confusing the field.

Therefore, we ask you to edit the text and submit the revised version. We decided to allow you one more round of revision because it will only require textual revision. With that said, all the reviewers expressed strong concerns that the current writing is not accurate enough, and if it is not fully taken care of in the next round, we must reject your manuscript, and no further revision will be allowed. In revising, we ask you to ensure that the revised main text is self-sufficient in conveying all the messages accurately and clearly, and not to utilize the rebuttal/response letter as a platform to explain things that are not entirely consistent with the main text.

Guidelines for revision:

Each Results section should clearly indicate how the data contributes (or doesn't) to the model. We also ask that in the discussion the authors recognize that there are observations they have made that are inconsistent with this model. They can do their best to try and explain them away, but at least readers will be able to more readily appreciate these inconsistencies and judge the author's explanations for themselves. Please ensure that the revised manuscript is self-sufficient in conveying these points, without the need of relying on response letter.

The reviewers consider that the most reasonable model to be put forward in this manuscript is "either Miranda BH-phospholipid or BH-actomyosin interactions are sufficient for cortical localization. In interphase BH-phospholipid interactions mediate uniform cortical association but early in prophase they start becoming inactivated by aPKC in an apical to basal fashion. By metaphase all BH-phospholipid interactions are inactivated and BH-actomyosin interactions take over (BH-phospholipid interaction might have supportive role in mitosis, too).

Overall, the reviewers agreed that there are many observations that do not exactly fit to the authors' model (e.g. overexpression of aPKC-deltaN). Some experiments were suggested to test authors' model, which resulted in observations that are inconsistent with the authors model. Each time, the authors provided 'possible explanations why the test did not support their model', yet remained that 'the model is still correct'. The authors should put more effort into explaining these inconsistent results (instead of 'explaining away' inconvenient results, trying to reach better explanation that makes sense as a whole). The reviewers do not expect the authors to figure out the mechanism of Miranda localization entirely. The reviewers appreciate that knowing that Miranda localization likely has two modes is sufficiently important progress, but writing is not conveying that this is the core message of the manuscript, and instead it claims more than the experimental data can support.

As an example of inaccurate statement, in the first section of the results, the authors start out by stating Miranda is 'cleared' and 'reappears' in mitosis, clearly leaving the impression that the previous model of expanding apical Par complex gradually displacing Miranda is wrong. However, as the video shows, Miranda 'clearance' occurs from the apical to basal. Although the authors explain that this clearing happens from apical to basal, but in subsequent sections, they do not come back to this fact and stick to the expression of 'clearance' and 'reappearance'. Here, the emphasis should be the fact that mitotic Miranda crescent is much stronger, prompting the investigation of the mechanism by which Miranda anchoring is promoted during mitosis (which they show to be actin dependent).

The weakest explanations in the current manuscript are the following:

1) If the authors' model is entirely correct, aPKC-deltaN expression should result in delocalization of Miranda in interphase, but normal, crescent localization in mitosis. But the actual observation is that aPKC-deltaN results in the delocalization of Miranda in interphase and mitosis. In the main text, the authors do not mention that Miranda fails to localize even in mitosis upon aPKC-deltaN expression, and conclude that their data is consistent with their model. In explaining this inconsistency, they state 'it is very difficult to predict the precise consequences of overexpressing a deregulated kinase'. A better explanation is required for the authors to be able to propose that their new model fits better than the existing model with all experimental results.

MiraS96D: the model predicts that MiraS96D should be cortically polarized in metaphase but it's partially cytoplasmic (S96D only partially destabilizes the BH-phospholipid interaction). The author's explanation is that cortical localization is affected more in interphase, which is not a thorough discussion. Unfortunately, readers will be wondering why Miranda5D (which completely abolishes BH-phospholipid interactions) wasn't tested because it would very easily resolve the explanations provided by the author for both of these inconsistencies (especially because the explanations are so poor).

2) According to the authors' model, Mira-deltaBH should be cytoplasmic in interphase, yet should localize to the basal crescent in mitosis. However, they find that Mira-deltaBH fails to localize to the crescent even in mitosis. In the main text, they simply conclude that 'this is consistent with Mira being localized to the cortex via interaction with plasma membrane'. Then, in response to the reviewers' comment that deltaBH shouldn't be cytoplasmic in mitosis, only in response letter, they point out that mitotic Miranda localization may require BH domain-mediated plasma membrane localization in addition to actomyosin. The authors should avoid any discrepancies between the main text and the response letter, and a cohesive story must be presented within the main text (and the observations that do not fit to the model must be clearly presented and acknowledged). Also, the reviewers do not agree that the data shows that Miranda disappears all at once upon LatA treatment. The revision should remove this argument and focus on the argument that Miranda disappears before aPKC arrives. Of course, we do not ask the authors to figure out everything about Mira localization, but they should explain 'why a particular observation betrays their prediction based on the model' in a cohesive manner, and more sincerely (i.e. do not bury important discussions within the response letter, which become inconsistent when placed in the main text).

We would like to provide an example of how reviewers' discussion proceeded. One reviewer noted: "I prefer an alternative interpretation, in which all of the localization after NEB is actomyosin dependent (ML-7 abolishes the crescent) and that this is mediated by the BH domain.": this clearly suggests that this reviewer appreciated the authors' discovery but still required to introduce his/her own interpretation to support authors' model. In response, another reviewer asked, "Then why is Miranda cortical in metaphase neuroblasts treated with LatA and lacking aPKC function (Figure 4—figure supplement 1 panel B)? If localization after NEB is actomyosin dependent, then surely it shouldn't localize after treatment with LatA." In response to this question, the first reviewer responded, "My interpretation of this experiment is that it indicates that aPKC-dependent clearing is an essential prerequisite for the formation of the basal crescent. In the absence of aPKC, Miranda remains in the interphase state where it binds to phospholipids. I could be wrong, as this result is entirely consistent with the simple (existing) model, but the latter doesn't explain the ML-7, Y26732 and FRAP data. What I find harder to understand is why the basal crescent doesn't form in the presence of constitutively-active aPKC, which is why the results aren't clear cut. It seems that Miranda needs to be phosphorylated by aPKC before NEB to form the basal crescent, but that too much aPKC activity or activity after NEB inhibits this. It is a shame that the manuscript didn't really get to grips with these questions."

In our opinion, a manuscript should not leave this much room of interpretation to the readers, and the authors must clearly present their data (or if that is not sufficient, more data would be clearly required. But the reviewers are not asking more experiments here).

Additionally, the model figure is currently extremely vague because it is hard to see what the authors are implying is taking place as far as Miranda-cortex interactions. Please revise the model figure to describe your model more clearly.

---

## [Author Response]

[Editors’ note: the author responses to the first round of peer review follow.]

Whereas all reviewers agreed that the idea of Miranda having two distinct modes of localization is interesting and of potential impact, they also felt that your data do not prove your hypothesis: whereas your data indicate that the existing model of Miranda localization may require a revision, your alternative hypothesis does not provide a better explanation than the current model, leaving unexplained observations as much as the current model does. The individual comments can be found at the bottom of this letter.The reviewers felt that it is critically important for you to carefully assess the current model and all the data that support it. At least the revised model must be better than the current one in explaining all the observations/data in the field. For example, a key result supporting the current model is that the unphosphorylatable Miranda (Mira5A) is depolarized in mitosis. This would be inconsistent with your revised model. In reaching a new model, it is not sufficient to point out that the current model is imperfect, but it is important that there is a significant advancement in the understanding, instead of muddying the field.Reviewer #1:[…]1) If Mira localization is regulated by two independent mechanisms, the significance of interphase cortical localization becomes unclear. The reason why Mira localization is important is because it functions as a fate determinant for NB daughter (GMC), thus only mitotic localization should matter.

We have not yet been able to identify a function for interphase Mira, However, as stated before, we have a working hypothesis. How are fate determinants prevented from acting in the NB? Very little is known about this. In mitosis, Mira shuttles between cytoplasm and basal crescent, therefore it is expected that not all of Mira (and presumably its cargos) segregate upon mitosis to daughter cells. Nevertheless NBs are sensitive to low doses of the Mira cargo Pros (Lau and Doe 2014). Keeping Mira at the interphase cortex might help to sequester Pros from entering the NB nucleus.

2) However, the Figure 4 data (Mira-deltaBH cannot go to mitotic basal cortex, whereas Mira-SD goes to basal crescent, despite not being able to localize to the interphase cortex) seems to suggest an interdependence between interphase and mitotic localization. For example, Mira has to localize to the cortex in interphase so that it can be phosphorylated by aPKC, then it dissociates from the cortex but potentiated for basal cortex localization.

We are not sure we fully understand this comment. While it is unknown where exactly aPKC can phosphorylate Mira in NBs, cytoplasmic Lgl might indeed restrict aPKC activity to the cortex. Therefore we agree, that it is a possibility that the localization of Mira at the cortex may allow phosphorylation by aPKC. The potentiation of basal cortex localization, once phosphorylated is also a possibility. The current data situation in NBs does not allow to be sure whether phosphorylation of Mira by aPKC has only as consequence the inhibition of cortical localization. It remains a possibility that phosphorylation of Mira enables it to engage with the actomyosin cortex basally.

Reviewer #2:This manuscript by Dr. Januschke and colleagues examined the mechanisms underlying the asymmetric localization of Miranda in the Drosophila larval neuroblasts. They proposed that there would be two different mechanisms that regulate the Miranda cortical localization; direct binding to the plasma membrane in interphase and actin-myosin dependent mechanism in mitotic phase. The concept that Miranda utilizes two different mechanisms to localize at the cell cortex in different cell cycle phases is new and interesting, however, there are some points to be clarified before published.Major comments1) In the most of the figures, authors utilized Lgl3A overexpression to abolish the function of aPKC, however, as shown in Figure 2, the majority of the Lgl3A expressing neuroblasts achieved asymmetric localization of Miranda in mitosis while in the aPKC mutant clones Miranda localized at the entire cortex (Figure 4), suggesting that Lgl3A overexpression does not recapitulate aPKC loss of function.

We have no reasons to doubt that overexpression of Lgl3A inhibits aPKC activity. As we stated before it is possible that overexpression of Lgl3A has different consequences than aPKC inhibition by other means. In addition we noticed that worniu-Gal4 is not expressed homogenously in all NBs (not shown). Therefore, Lgl3A levels are likely to vary between individual NBs. We interpret the different effects on Mira by Lgl3A overexpression to mean that aPKC is inhibited to different degrees.

Although Lgl3A is thought to suppress aPKC function, it seems that, at least in these cases, Lgl3A expression is not enough to suppress aPKC function or it has some other functions than aPKC suppression, making it difficult to interpret Lgl3A overexpression phenotypes. Thus, it is crucial to check what is observed in Lgl3A overexpression is also the case in aPKC mutant or RNAi clones.

Failure to clear Mira from the interphase cortex has been observed in mitotic apkc NBs and actin network dependence of Mira localization was tested in aPKC RNAi NBs, which we confirm to have reduced aPKC levels (Figure 2—figure supplement 1). Therefore, the only experiment that solely relied on Lgl3A overexpression is the analysis of Mira cortical dynamics by FRAP (Figure 3). We have now performed FRAP of basal Mira in mitotic aPKC RNAi NBs and find it to be similar to when Lgl3A is overexpressed (new Figure 3’).

We do see enlarged Mira crescents in a few NBs when aPKC RNAi is driven by worniu-Gal4 (not shown) in line with the Lgl3A overexpression results. Whatever the actual effect on aPKC, the importance of this experiment lays in the fact that disrupting the actin cortex, results in loss of basal enrichment of Mira. Importantly, we see a similar effect of LatA treatment on Mira^S96D^ in which aPKC activity in principle should be normal unless Mira^S96D^ has an inhibitory effect on aPKC, which seems unlikely.

2) In Figure 4, the authors showed that Mira[S96D] showed uniform cortical localization in colcemid and LatA treated neuroblast. Following the authors' model, Miranda localizes at the cortex by either direct binding to the plasma membrane or actin-dependent mechanism, however, in this situation Mira seems to localize at cell cortex in the absence of both machinery. Then how does Miranda localize at cell cortex in this situation? Authors should discuss about this point.

We agree that this is a highly unexpected observation. In our view the phosphomimetic Mira^S96D^ demonstrates the point that the way Mira localizes to the cortex in interphase (weak interaction with phospholipids drives plasma membrane localization) is different to that in mitosis (Mira is stabilized additionally by the actomyosin cortex basally). Even though Mira^S96D^ localizes uniformly to the PM upon microtubule and actin network disruption, it does not do that when the actomyosin cortex is intact. Mira^S96D^ is therefore able to read the information required for its asymmetry and stabilized basally.

The Aspartate replacing the Serine residue at position 96 only mimics phosphorylation and as is true for any phosphomimetic mutant, the ability to regulate the protein at the modified residue is gone. Therefore uniform plasma membrane binding upon LatA treatment could be an irrelevant consequence of the Mira^S96D^ mutant. Alternatively, general aspects of Mira (ability to dimerize, other posttranslational modifications (Jia et al., 2015, Slack et al., 2007, Zhang et al., 2015)) might change upon NEB and contribute to Mira’s ability to localize to the PM.

3) Authors utilized Y-27632 to temporally inhibit aPKC function but it is not clear whether Y-27632 treatment is really affecting the aPKC activity or not. As a control experiment, it is better to examine whether the addition of 50uM Y-27632 to the cycling neuroblasts results in the Lat-A-insensitive entire cortical localization of Miranda in mitotic phase (like aPKC loss of function situation).

We have now performed the requested experiment. We added 50, 100 and 200µM Y-27632 to cycling NBs co-expressing Baz::GFP or aPKC::GFP with Mira::mCherry. 50 and 100µM resulted again in enlarged crescents. Only with 200µM we were able to detect uniform cortical Mira in mitotic NBs. When 200µM were added to cycling NBs, Baz::GFP started to look abnormal in interphase and mitosis and so did NB cell shape in interphase. However 8 out of 25 NBs in this condition divided, 7 of which had incomplete clearing of Mira at the onset of prophase and Mira was everywhere on the cortex upon NEB. When such NBs divided the daughter cells appeared similar in size (new MOV S15). We repeated this experiment co-expressing aPKC::GFP and Mira::mCherry, but arrested the NBs with colcemid in mitosis. We were able to detect in few cases apical aPKC crescents and uniform cortical Mira. Adding LatA to these cells, lead to loss of cortical aPKC asymmetry, but Mira remained at the cortex (new Figure 6).

These experiments are in agreement with the idea that Mira phosphorylation by aPKC is not occurring in the presence of high doses of Y-27632.

Reviewer #3:This revision, like the original, proposes that Miranda is polarized in Drosophila neuroblasts via two distinct mechanisms, one in interphase and one in mitosis – the former being by a previously described "BH" association mechanism, and the latter being a "myosin affinity zone". I had two main criticisms of this proposed mechanism. First, the authors did not separate these two functions, rather deletion of the BH caused loss of Miranda polarity in both interphase and mitosis. Second, the proposal of a "myosin affinity zone" is based on data using inhibitors leaving open the possibility that the effect is indirect. The author's work is potentially exciting because there are interesting questions related to Miranda's function in interphase, and the role of myosin in polarizing Miranda. Unfortunately, however, I do not believe that the revised manuscript comes close to adequately addressing the criticisms raised in the original review and therefore I do not recommend publication.• A separation of function allele would be the most definitive test of the authors model – if two different mechanisms are used, then it should be possible to find Miranda alleles that only polarize in mitosis and not in interphase (unless the interphase mechanism is required for the mitotic one, which is actually what the title says – see below). Mira∆BH isn't polarized in interphase or mitosis indicating that the authors model is wrong or the BH motif is required for each binding mode. To distinguish between these possibilities, the authors examined MiraS96D, a phosphomimetic allele, and conclude that it is a bona fide separation of function allele, disrupting interphase localization but not mitotic. However, S96D is clearly a hypomorph (Figure 2 Current Biology 19, 723-729, 2009) and my interpretation of the author's data (i.e. Video 9) is consistent with it behaving as a hypomorph in their assay – Mira96D basal crescent signal is reduced and spindle signal is increased not only in interphase but also in mitosis.

Indeed elements in the Mira protein mediating the interphase mechanism are also required for the mitotic mechanisms, but the mechanism mediating binding in interphase is not sufficient in mitosis. That is what we propose.

We take the criticism that we apparently failed to be clear. But we respectfully would like to point out that we never argued that the BH motif is not required in mitosis nor did we claim that Mira^S96D^ is not able to bind the PM, we also fully agree that *mira^S96D^*is a hypomorph with regards to efficient PM binding.

However, the negative charge within the BH motif strongly reduces PM binding in interphase, but Mira^S96D^ is always asymmetric in mitosis. This is an argument that the affinity of Mira for the cortex changes in mitosis and hence that the mechanisms of cortical retention in both cases are different. Somehow this has been perceived as if we were to use this experiment to rule out a role of the BH motif in mitosis. We have improved this point in the current version to avoid further misunderstanding.

Furthermore we would like to point out that additional features of Mira protein are altered when the entire BH motif is deleted. Mira^S96D^ can bind microtubules and PM in interphase and in mitosis. Mira^ΔBH^, however, is not enriched at the cortex in interphase or mitosis. Furthermore, unlike Mira^S96D^, Mira^ΔBH^ apparently does not bind spindle microtubules in metaphase. Therefore, the deletion of the entire BH motif affects other properties of Mira than just mediating binding to the PM.

• I also criticized the original version of the manuscript because it proposed that mitotic Miranda polarization occurs through a "myosin anchor" in a "basal affinity zone" because the data supporting this relies on drug treatments. Nothing in the revised manuscript addresses this criticism. The author's rebuttal letter states, "We agree that the mode of action of myosins remains perhaps obscure since we do not know whether the effect is direct or indirect." That is precisely my criticism, that the mode of action of myosins remains obscure. The letter goes on to state, "However, given that the ML-7 effect on Mira basal localization in colcemid arrested NBs can be tempered by overexpressing a phospho-mimetic version of Sqh[…]an involvement of myosin activity in anchoring Mira at the basal pole at NEB seems very likely." This statement is not very impactful if the involvement of myosin activity is very indirect, which the authors themselves acknowledge is entirely possible.

We agree that the data is based on drug treatments and therefore weaker than the genetic evidence for a role of myosin activity previously provided (Barros et al., 2003). Our experiments allow narrowing down the function of myosin activity to either provide anchoring for Mira or contribution to configure the basal affinity zone. The integration of phosphoregulation and the regulation of the actomyosin network is key to understand how Mira localizes asymmetrically.

The letter further states, "We hope that the reviewer would agree that the current apical exclusion model does not leave much room to explain the role of myosin activity in Mira localization[…]". I strongly disagree with this statement. First, pretty much any model would allow for an indirect role for the cytoskeleton. But more importantly, the authors do not seem to understand the "current apical exclusion model", at least as it is articulated in Bailey and Prehoda, Dev Cell, 2015, which they have referenced. In this model, BH motifs cooperate with accessory interactions to mediate cortical recruitment of Par substrates (e.g. see Discussion section, "Multivalent Interactions Mediate Par Substrate Cortical Localization"), and that aPKC phosphorylation of the BH motif disrupts this interaction. In this model an "accessory interaction" could certainly require interactions with myosin – the key point being that any such interaction must not be sufficient for cortical targeting (otherwise BH phosphorylation wouldn't have an effect on cortical localization).

We thank the reviewer for pointing this out. The central idea of Bailey and Prehoda, 2015, that is based on experimental evidence, builds on experiments using overexpression of different versions of Mira (and other substrates) in S2 cells and in vitro assays to elegantly establish that phosphorylation of the BH motif by aPKC inhibits the ability of aPKC substrates to bind to the plasma membrane. If anything our data confirm this mechanism now in NBs using reporters and tools at endogenous levels of expression: Mira weakly associates with the PM in interphase, which i) depends on the BH motif, ii) is perturbed by introducing a negative charge at an aPKC phosphorylation site within the BH motif and iii) prevented by overexpressing constitutively active aPKC.

One testable interpretation of the data provided in (Atwood and Prehoda, 2009) is that controlling where aPKC is active in mitotic NBs, provides the spatial information where Mira can bind to the cortex. If this were the case, once aPKC is activated at prophase onset, Mira should be removed apically, but spared basally. This is not what we find. Moreover, the way Mira interacts with the cortex in mitosis should be the same regardless of the state of aPKC activation. This is again not what we find. Our data rather suggest that at least in the case of NBs additional processes occur during NB polarization which provide critical spatial information for Mira localization. Thus, our proposal of a basal affinity zone is in perfect agreement with the discussion about multivalent interactions in Bailey and Prehoda, 2015.

However, rather then being an indirect bystander, we propose that the actomyosin cortex directly provides spatial control of asymmetric Mira localization, which we believe is linked to control of asymmetric daughter cell size.

• The title of the paper is incorrect, according to the authors. The title states, "Switching the cortical binding mode is required for asymmetric fate determinant localization in Drosophila neuroblasts". However, the authors claim to have a Miranda allele that doesn't switch cortical binding modes and is still asymmetrically localized, directly refuting the statement made in the title. Stating one thing in the title and another in the text will confuse readers and make it very difficult for those in and outside the field to understand the impact of the paper. Furthermore, it suggests that the authors themselves do not understand the impact of their work.

We reveal that the cortical Mira binding mode can be distinguished in interphase and mitosis based on the actin dependence, altered cortical dynamics and sensitivity to introducing a negative charge into the BH motif. At one point a change in the mode retaining Mira must occur. We felt that “switch” described this well. However, this has apparently been misleading and perceived as if we were to exclude a role of the BH motif for localization in mitosis, which as stated already previously and above is not what we want to put forward. We have therefore taken the advice and changed the title to “aPKC-mediated displacement and actomyosin-mediated retention polarize Miranda in *Drosophila* neuroblasts”. If aPKC does not phosphorylate Mira, it remains uniformly bound to the PM. After NEB, Mira is however able to read spatial information provided by the actomyosin network to asymmetrically localize. Therefore aPKC-mediated cortical displacement is necessary but not sufficient to asymmetrically localize Mira asymmetrically.

[Editors' note: the author responses to the re-review follow.]

Reviewer #1:This is a revised version of previously revised manuscript by Januschke and colleagues. They report the molecular mechanism of asymmetric Miranda localization in Drosophila neuroblast.Same as the previous versions of this manuscript, I found that each experiment is well done and well documented. However, the writing and presentation again makes it difficult for me to assess general impact of this manuscript (thus I have to rely on other reviewers with more expertise in the field). The main difficulty in reading this manuscript for me was 1) sometimes it is not clear whether all of their conclusions are consistent among themselves, 2) sometimes it is not clear whether their claims are indeed consistent with previous findings especially when they argue against existing models. (examples will be provided below). Here, I am not trying to be nit-picky about writing, but the lack of clarity throughout the text has made it difficult for me (and probably for other reviewers in the previous rounds) to grasp the (potential) impact of this paper. With that being said, I am not saying that writing only can be a reason to accept or reject a paper.In addition, in the course of revisions, the authors often responded to reviewer comments only in their rebuttal/response letters, without incorporating it into the main text. This has also made it difficult to relate how the concern was resolved in a way presentable to the future readers. Response letters should be in the format of 1. Reviewer comments, 2. response by authors, 3. Explanation how the authors incorporated reviewer comments and responses within the main text ("based on these.[…] we changed the main text as following […]"). In the previous versions, all the reviewers understood that the manuscript clearly demonstrates that the current model needs to be revised. However, the reviewers also commented that the authors' new model may not be consistent with the existing data. In response, the authors provided explanations only in the response letter, without changing much in the main text. The authors should keep in mind that the future readers may have the same questions as reviewers (very likely), and thus such questions from reviewers are better be addressed within the main text, not in the rebuttal letter.

We followed the advice of the editors and the reviewers to radically revise the manuscript.

Reviewer #2:This manuscript contains a series of well-controlled and careful experiments that convincingly demonstrate that Miranda localises by different mechanisms in interphase and mitosis. The observations that Miranda is excluded from the cortex by aPKC and retained in a basal crescent by the acto-myosin cortex are not novel, but most previous work has presented these as competing models to explain how Miranda is asymmetrically localised. Thus, the main novelty in this manuscript is that both are true, but at different time points during the cell cycle. While I agree with the other referee's comments, I don't think that any of them challenge this basic conclusion, which in my opinion is sufficiently important to merit publication in eLife.While I am very positive about these results and the quality of the data, I think that the presentation needs to be improved. My version was entirely lacking Figure 3 (which contains one of the key pieces of data showing that Miranda behaves differently in mitosis), and the videos were not labelled in the same way for downloading as in the text, which made it difficult to relate the two. The text also needs to be improved to highlight the main conclusions and discuss how these relate to previous work. To give just one example, the manuscript presents strong evidence that myosin activity is required for the basal recruitment of Miranda after NEBD (ML-7 treatment, rescue of this effect with SqhEE, low doses of Y-27632), but does not discuss these data in light of previous work from Barros et al. arguing that myosin excludes Miranda from the cortex or the argument against Barros et al. that high concentrations of Y-27632 inhibit aPKC. Januschke et al. seem to have cleared up all of this confusion by using a different drug, proving that it is specific for myosin activation and by separating the effects of Y-27632 on Rho kinase and aPKC at different doses, but they fail to put this all together into a single argument in the discussion that makes their contribution clear.

We have substantially changed the entire manuscript following the advise of the reviewer. To highlight our contribution we have specifically modified the first paragraph of the Discussion, which summarizes the relevance of the problem, our approach, and the main conclusion. Our discussion now dissects the precise contribution of aPKC and actomyosin along our data and the literature. The discussion of our data in light of the Barros et al., 2003 data and the Atwood and Prehoda, 2009 proposal is discussed in the Discussion section.

Reviewer #3:This manuscript by Januschke's group proposed that the asymmetric Miranda localization in the asymmetric cell division of the Drosophila neuroblast is mediated by two mechanisms: cortical exclusion through the phospho-regulation by aPKC and cortical retention by actomyosin network. The concept that Mira utilizes two different mechanisms in different mitotic phases is very interesting but their data does not seem to fully support their idea and seem to leave some room for alternative interpretations.First of all, there was no Figure 3 in this manuscript.I think we cannot accept such incomplete manuscript for a peer-reviewing process, but I at least leave some comments.In the Lgl3A over-expression experiments, as authors mentioned in the letter, the degree of the aPKC suppression seems to be varied; 7/25 seem to completely suppress aPKC and 4/25 seem to fail to suppress aPKC function. I think it is natural to treat the rest 14/25 as the intermediate between these two; a partial suppression of aPKC function. In such situation, " Mira initially asymmetrically localized, but LatA treatment cause Mira to become uniformly localized on the PM". This result is mostly same as the result when authors introduced MiraS96D in which " MiraS96D::mCherry always achieves asymmetric localization in mitosis…MiraS96D::mCherry relocalizes to the entire cortex upon LatA treatment".Thus, it is highly possible that MiraS96D is a hypomorphic allele with regards to the phospho-regulation by aPKC.In this point of view, I think these results would not support the authors' idea that " the BH motif mediates two different binding modes", and rather it is possible to interpret that Mira in mitosis would require both PM binding and other supportive interactions mediated by actomyosin for basal localization.

To improve focus, we removed the experiments combining Lgl3A and LatA as well as S96D and LatA and focus on the point that in interphase Serin96 is a critical residue involved in PM binding of Mira.

We now further include the phosphomutant S96A, which is not cleared, and even transiently localizes apically before NEB after which it localizes uniformly at the cortex in an F-actin independent manner (new Figure 4). In the S96D mutant the Aspartic Acid at position 96 strongly removes uniform binding to the plasma membrane in interphase, driving Mira onto cortical microtubules (new Figure 4). Therefore, this shows that S96 phosphorylation is critical for clearance from the PM.

As we stated before this does not exclude the possibility that the BH motif is required for both. Our new results treating colcemid arrested NBs with 200µM Y-27632 reveal that despite occurring after a significant delay (~50min) Mira accumulates apically when 200µM Y-27632 is added to colcemid arrested NBs (new Figure 5). Therefore, while the precise contribution of aPKC after NEB, remains to be determined, it is possible that aPKC contributes to Mira asymmetry after NEB by displacing Mira from the apical cortex as suggested (Atwood and Prehoda, 2009). However, at this concentration other effects of Y-27632 cannot be ruled out. This is discussed in the Discussion section.

It appears to be clear that Mira localization occurs step-wise. It is first uniformly binding the plasma membrane, which is prevented by aPKC phosphorylation. Then after NEB (in agreement with other findings (Zhang et al., 2016). Mira reappears at the cortex. The BH motif is required for this to happen. We recently found that Mira interaction with its cognate mRNA is required to maintain its localization basally (Ramat et al. *in press*, uploaded for reviewers). These results support the notion, that the BH motif is required for Mira asymmetry establishment, which allows subsequent stabilization through a mechanism requiring actomyosin activity.

In the most of the figures, authors used Baz-GFP as a marker for the Par complex localization. Baz localization and aPKC localization are not always the same as shown in Figure 6.

(This point was also raised by reviewer 4). The new Video S2 shows an aPKC::GFP Mira::mCherry expressing neuroblast, in which Mira dynamics in the transition from interphase to mitosis are similar if not identical to when Mira::mCherry is expressed alongside Baz::GFP. While the cytoplasmic levels of Baz::GFP and aPKC::GFP may vary, especially after incubation of NBs with multiple drugs, we have no evidence for differences in the cortical distribution of aPKC and Baz once NBs enter mitosis.

Thus, it would be important to examine the distribution of aPKC itself. Especially, it is important to examine the kinetics of aPKC in the presence of LatA (Figure 2), because this is the only one result that distinguishing whether Mira was "cleared" by aPKC or "anchored" by actin.

We would like to point out, that regardless of the Par complex marker that is co-expressed, Mira crescents fall off homogenously from the cortex in colcemid arrested NBs when LatA is added. This is measured in Figure 2—figure supplement 1 of the revised manuscript: the ratio of fluorescence intensities found in a region of interest (ROI) at the extremity of the crescent and in a ROI in the centre remains constant. Taking the effect on uniform cortical Mira (displacement into the cytoplasm in an apico basal direction), which is likely to be driven by aPKC at the onset of mitosis as reference for cortical Mira behaviour in response to aPKC, the effect of LatA on Mira in colcemid arrested NBs is different as we do not observe directional (i.e. apical to basal) removal of Mira.

Nevertheless, following the advice of the reviewers 3 and 4, new Figure 2—figure supplement 1, now includes the kymograph analysis of the effect of LatA on colcemid arrested NBs that express Mira::mCherry and functional aPKC::GFP (Besson et al., 2015, new Video S4), which reveals the same picture as when using Baz::GFP as a marker for the Par complex. Plotting the cortical fluorescence intensities of both Mira and aPKC reporters revealed that Mira is lost 2.8min ± 1min (n=13, this information is found in the Results section), before aPKC levels rise above cytoplasmic levels at the basal pole.

Both these findings support the interpretation that anchoring to F-actin is critical to keep Mira basally rather than an indirect effect caused by cortical aPKC redistribution.

It is difficult to understand the different outcomes of the ML-7 treatment and Y-27362 treatment. Myosin inhibition resulted in the displacement of Mira from the cortex and Myosin/actin inhibition resulted in the expansion of Mira crescent, why? Authors should mention their interpretation about these results.

This is an interesting point to discuss. The regulation of myosin activity by phosphorylation of its regulatory light chain is complex. Studies in mammalian cells have revealed that phosphorylation of myosin regulatory light chain can be differently affected by Y-27632 and ML-7, which could result in different outcomes on myosin activity (Watanabe et al., 2007). We have added a section in the Discussion addressing this point.

Reviewer #4:This manuscript proposes a mechanism for the polarization of the fate determinant Miranda in Drosophila neuroblasts. The current model for this process is that Miranda directly interacts with the plasma membrane and phosphorylation of the membrane-interacting peptide by aPKC displaces Miranda into the cytoplasm. The model proposed in the work under consideration is that Miranda polarization occurs via different mechanisms in interphase and mitosis, and the mitosic mechanism involves a "basal affinity zone" that somehow involves the cytoskeleton. In general, I feel that the authors have overinterpreted their data, which relies on drug treatments and in some cases, used flawed logic to draw their conclusions. Some of the main problems are summarized below:• While the characteristics of interphase localization are interesting, they are not significant unless interphase localization is somehow important for Miranda function

Reviewer 1 has raised this point as well. New Figure 1—figure supplement 1 shows staining of larval NBs in interphase and mitosis for Mira and Pros. Pros, like Mira, is uniformly at the NB interphase cortex in the larval brain. We further stained *mira* mutant NBs in the same way and reveal that Pros is found in the nucleus. This is in agreement with previous findings in embryonic NBs (Matsuzaki et al., 1998). Therefore it is a strong possibility the PM-bound Mira in interphase is linked to regulating nuclear Pros (see main text: Introduction, Results and last paragraph of the Discussion).

• Drug treatments, such as LatA, lead to loss of Miranda cortical localization, which could result from drug-induced ectopic aPKC activity. The experiment the authors provide to discount this possibility ("anchoring vs. clearing") is not convincing for a number of reasons, including that Baz localization is used as a proxy for aPKC localization. The cortical localization of Miranda in LatA treated neuroblasts lacking aPKC function are consistent with a misinterpretation in this regard.

Following the advice of reviewers 3 and 4, new Figure 2—figure supplement 1, now includes the kymograph analysis of the effect of LatA on colcemid arrested NBs that express Mira::mCherry and functional aPKC::GFP (Besson et al., Curr Biol 2015, and new Video S4), which reveals the same picture as when using Baz::GFP as a marker for the Par complex. Plotting the cortical fluorescence intensities of both Mira and aPKC reporters, further revealed that Mira is lost 2.8min ± 1min (n=13), before aPKC levels rise above cytoplasmic levels at the basal pole. This information is found in the Results section.

Furthermore, LatA treatment results in apical to basal redistribution of aPKC. Nevertheless, Mira crescents fall off homogenously from the cortex. This is measured in Figure 2—figure supplement 1: the ratio of fluorescence intensities found in a region of interest (ROI) at the extremity of the crescent and in a ROI in the centre remains constant. Taking the effect of aPKC on uniform cortical Mira, which is driven by aPKC phosphorylation in an apical to basal manner at the onset of mitosis as reference, the effect of LatA on Mira is different, as we do not observe “ends-on” reduction of crescents. Both these findings support the interpretation that anchoring to F-actin is critical to keep Mira anchored basally rather than an indirect effect caused by cortical aPKC redistribution.

The new Video S2 further shows an aPKC::GFP Mira::mCherry expressing neuroblast, in which Mira dynamics in the transition from interphase to mitosis are similar if not identical to when Mira::mCherry is expressed alongside Baz::GFP. While the cytoplasmic levels of Baz::GFP and aPKC::GFP may vary, we have no evidence for differences in the cortical distribution of aPKC and Baz once NBs enter mitosis.

• The authors over interpret the S96D mutant, which only partially abrogates plasma membrane binding. They state in the text that the localization of mutant is the same as WT but the videos they provided do not support this claim.

Following the advise of this reviewer, we revised this part of the manuscript. We now include the S96A mutant, which we find not to be cleared at prophase onset, but to transiently accumulate in the wrong place, i.e. apically before NEB, after which it localizes uniformly in an actin-independent manner in mitosis (new Figure 4). These results show that mutating only one of the identified aPKC phosphorylation sites is sufficient to prevent cytoplasmic displacement of Mira at the onset of mitosis.

When this site is mutated to Aspartic Acid (S96D), unlike wild-type Mira, S96D localizes predominantly to cortical microtubules. However, in mitosis S96D, is always able to localize asymmetrically, which we fully agree may occur at reduced levels.

Following the advice of the reviewer we have revamped the paragraph and figure showing the S96D results, emphasizing the point that interphase and mitotic localization are differently affected by the S96D mutant, supporting the point that Mira binds to the plasma membrane in.

• Even if the previous problems are overlooked, the authors have not gone far enough to characterize the new interactions that they are proposing take place. The phenotype they observe when inactivating the cytoskeleton could arise from very indirect effects, yet they are proposing direct interactions with a "basal affinity zone". I do not believe that they have presented data supporting their claim that the zone is only basal, or that tell us anything about the interactions that would occur in the zone.

Following the advice of the reviewer we do not refer to basal affinity zone any more and more generally discuss the role of actomyosin remodeling.

• The text is poorly-written and convoluted. It would be very difficult to follow for someone outside the field.

We have radically revised the manuscript, summary, Results and Discussion, and removed data (Lgl3A overexpression in combination with LatA treatment) to improve focus and clarity.

[Editors' note: the author responses to the re-review follow.]

Reviewer #1:[…]However, the major weakness of this manuscript in writing remains. Aside from multiple clear grammatical mistakes, the lack of contexts when they describe their data really interferes readers' understanding (the most important question in the field that is addressed by each specific experiment is not explained). I listed up several specific places, where the way it's written confused me a lot. I recommend the authors to go through extensive editing focusing on 'reader friendly writing'. Given the history of this submission, we must insist that the next version will be the last. Please be certain that you have had others read and proof this so that the Board and reviewers will have an easier time deciding if you have made your essential points. Failing that, the paper will not be returned for another rounds of revisions.

We have revised the way the manuscript is written substantially following the reviewers advice. We have consulted a professional editing service to ensure grammatical correctness and to improve reader friendly writing. In addition, we had several scientific colleagues in the division and the school of life sciences distant to the field to read and comment on our revised version. We especially put an emphasis on explaining the rationale behind each experiment and hope that this has improved the clarity of the paper.

Reviewer #2:This manuscript contains a series of well-controlled and careful experiments that present evidence that Miranda localises by different mechanisms in interphase and mitosis. The observations that Miranda is excluded from the cortex by aPKC and retained in a basal crescent by the acto-myosin cortex are not novel, but most previous work has presented these as competing models to explain how Miranda is asymmetrically localised. Thus, the main novelty in this manuscript is that both may be true, but at different time points during the cell cycle. The experiments are carefully quantified and informative, but more care is needed in the interpretations of the results. The authors should discuss alternative explanations for their data, even if they are not their preferred model. Another obvious weakness is the use of Y-27632 as an aPKC inhibitor when it also affects myosin activity through ROCK, although this experiment is still valuable and provides a useful comparison with previous work. The manuscript is an improvement over the previous version, but needs to be revised to include a more comprehensive discussion of the meaning of the results that considers alternative interpretations and that is more accessible to a general audience.

Following the advice of this and the other reviewers, we have completely restructured the Results section detailing the Y-27632 experiments. We have split this paragraph into two sections to improve clarity, especially to highlight the fact that while high doses of Y-27632 (200µM) inhibit aPKC, lower doses (25µM) of Y-27632 seem to affect Mira crescent size, which is likely to occur independently of aPKC inhibition. We have taken a new angle at the discussion following the advice discussing alternative interpretations to explain Mira localization. This includes discussing the role of known inhibitors (e.g. Lgl, Betschinger et al., 2003, Atwood and Prehoda, 2009), changes in the spatial regulation of aPKC based on recent observations made in the *C.elegans* zygote (Rodrigues et al., 2017, Wang et al., 2017) as well as general changes in Mira protein upon nuclear envelop breakdown such as dimerization (Yousef et al., 2008, Jia et al., 2015). We have expanded further on how we think actomyosin might affect Mira localization, to explain our findings better. We feel that the new angle the discussion now takes improves clarity and hope that it explains better the general relevance of our results to a broader readership.

Specific comments:1) FRAP stands for Fluorescence Recovery after Photobleaching and not Fluorescence Redistribution.

We have change this as advised.

2) Intriguingly, such enlarged crescents were LatA and ML-7 sensitive, both indicators that enlarged Mira crescents were not due to lack of phosphorylation of Mira by aPKC. This needs more explanation and perhaps some discussion as to whether the expansion depends on aPKC phosphorylation of another substrate.

We have explained further this section of the Results, which now includes additional clarification: “Furthermore, the enlarged Mira crescents resulting from Y-27632 addition to cycling NBs were sensitive to LatA and ML-7 treatment, as were normal Mira crescents in controls. This suggests that also under this condition, actomyosin is important to retain Mira at the cortex mitosis, even when the size of the crescents is enlarged (Figure 5; see Figure 5 for Mira crescent size quantification under the different conditions).”

Given that we do not have any data of other substrates of aPKC that are linked to actomyosin regulation, although a tantalizing possibility, we have decided not to expand the discussion on how that might affect altered Mira crescent size. However, aPKC is likely to contribute after NEB to Mira asymmetry by continuing to remove it from the plasma membrane apically. This was revealed with higher doses of Y-27632, which phenocopies Mira localization defects seen in *apkc* mutants, but given the nonspecific nature of Y-27632 this cannot be precisely determined in this way. This is stated in the Results section.

Reviewer #3:This revision from Hannaford et al. remains seriously flawed and I do not recommend it for publication. First, many of the author's conclusions are contradicted by their own data. For example, a key conclusion is that Miranda is cleared from the cortex at prophase. It's difficult or impossible to make this assessment from the data the authors show because of Miranda signal from attached GMCs. Video S2 (included for a different reason) shows a beautiful division without any visible GMCs. This video supports a conclusion opposite of the author's: Miranda remains on the cortex throughout the cell cycle including in prophase.

We thank the reviewer for pointing this out. The key starting observation for our study relates indeed to the cortical dynamics of Mira as the NBs transits from interphase into mitosis. We find that just before NEB Mira has been removed from certain regions of the cortex, to which Mira returns after NEB to form the basal crescent. However, we did not and still do not claim that Mira is completely lost from the basal cortex just before NEB. We now made sure that this result and conclusion is clear in revised relevant sections. What we found is that there is a difference in quality and quantity of Mira localization at the basal NB pole just before and after NEB. This is quantified in Figure 1’ and visible in Video S1 (compare frame 00:54 and 01:00), Video S2 (compare frame 01:04 and 01:12) as well as Video S8 (compare frame 00:39 and 00:45). If anything, Mira signal from GMCs would mask the extent to which Mira was removed from the NB cortex, bearing the risk to under- rather than overestimate the extent to which Mira is removed from the basal cortex just before nuclear envelope breakdown.

Another example: if Miranda is cleared all at once from the cortex upon LatA treatment, as the authors conclude, then why does the kymograph show the Miranda crescent shrinking? This effect is obfuscated by compressing the transition over a small number of pixels but Video S4 shows it beautifully. Scrubbing video S4 back and forth around the time that Miranda disappears clearly shows the crescent growing and shrinking, not disappearing all at once.

Fluorescent levels of Mira are low since we use the endogenous promoter to drive Mira::mCherry expression. Therefore, variations in signal-to-noise may lead to variations in detecting Mira between samples. To illustrate this point, we provide now additionally Video S5. In this experiment Baz::GFP and Mira::mCherry co-expressing colcemid-arrested metaphase NBs were treated with LatA and the fluorescence levels on the cortex are plotted for both markers. Like Video S4 this video also shows that Mira becomes cytoplasmic before changes in Baz (or aPKC) can be detected at the cortex. Importantly, in new Video S5, Mira crescents seem to broaden before homogenously fainting at the basal cortex (compare frame 00:00 to 05:45). In any case, we have not based our conclusion on individual samples, but measured the response of Mira cortical fluorescence to LatA treatment across several samples, the analysis of which was done on the raw datasets. In the kymographs and the fluorescence intensity measurements at the margins of Mira crescents and in the center are shown in Figure 2—figure supplement 1. Therefore, taken the behaviour of Mira on the interphase cortex in response to mitotic entry as a reference, when Mira is cleared in an apico-basal manner, our quantitative data indicates that Mira falls off homogenously from the cortex supporting a role for actomyosin in retaining it basally after nuclear envelope breakdown.

Another example: The MLCK inhibitor ML-7 causes cortical Miranda to be lost in metaphase, but the phenotype could be non-specific (SqhEE expression only causes delay of the phenotype), and even if it isn't this one result doesn't prove that Miranda loss is due to a direct effect. Thus, close inspection of the author's data does not support clearing of Miranda at prophase or loss of anchoring by cytoskeletal poisons (in fact, for two experiments the opposite conclusion is supported).

We acknowledge that small molecule inhibitors can have off target effects, but these can be mitigated to some extent by using several that act in different ways to perturb a pathway (i.e. LatA and ML7) and assessing if both have the same effect. Rescuing the effect Y-27632 or ML-7 by expressing a phosphomimetic version of the substrate is a further well-established approach (e.g. Das and Storey, 2014, Barros et al., 2003 and Bertet et al., 2004) to test the specificity of effects. Why we only see a significant delay with this approach and not complete suppression, might be due to the levels of the phosphomimetic protein supplied, which are difficult to quantitatively control. We agree that this experiment does not show a direct effect on Mira, neither does it rule that out, it could well be caused indirectly through changes in the actomyosin network. We have elaborated on how we think actomyosin activity might contribute to stabilize Mira localization in the discussion, which accommodate also indirect effects, such as changes in the local tension landscape. This can be found in the new Discussion.

The paper contains a fair amount of data, such as the difference in photobleaching dynamics of Miranda in interphase and mitosis, and the examination of Y-27632 treated neuroblasts, which appear to be superfluous without a more clear explanation.

The photobleaching experiments are important as they lend strong support to the idea, that the way Mira behaves at the cortex, i.e. its turnover in interphase and mitosis are different. The results obtained by measuring cortical Mira dynamics by FRAP, therefore support our idea that Mira retention at the cortex has at least two states: BH motif mediate retention at the plasma membrane (interphase) and actomyosin-dependent Mira retention/stabilization at the basal cortex after nuclear envelope breakdown. We have revised this section and emphasized this rational before detailing the Results (“Second, the turnover of Mira at the cortex when bound to the PM in interphase and when interacting with F-actin after NEB should be different.”).

As stated in the replies to the other reviewers comments, we have revised the section describing the Y-2763 experiments to improve clarity and provide more information and elaborated on the interpretation of these results in the Discussion as mentioned above.

Considering that the paper's foundational conclusions are flawed, it is not surprising that the strongest tests of the resulting model (that Miranda is cleared from the cortex by aPKC phosphorylation of its BH motif during prophase and anchored by actomyosin during metaphase) fail miserably. These tests come in three experiments. First, Miranda∆BH – if the BH is only required for interphase localization, then Miranda∆BH should be cytoplasmic and interphase and polarized during metaphase. However, the authors find that it's cytoplasmic in metaphase too. Expression of constitutively active aPKC (aPKC ∆N) is also a great test of the model. If aPKC activity is just required to clear interphase cortical localization, constitutively active aPKC should result in cytoplasmic Miranda in interphase and polarized Miranda in metaphase. The result again contradicts the model's prediction: Miranda is cytoplasmic in both.

We would like to point out, that the mechanism that we are proposing is compatible with a potential role for the BH motif in interphase and in mitosis. The point that we are trying to make is that while plasma membrane binding is sufficient to retain Mira at the membrane in interphase, the BH motif dependent mechanism plus an additional actomyosin dependent process is required to stabilize or retain Mira at the cortex after nuclear envelope breakdown. This view is consistent with propositions found in the literature (e.g. Bailey and Prehoda, 2015).

An interesting side observation is further that when the BH motif is deleted, Mira is not found on mitotic microtubules (Figure 3) as wild type Mira protein typically is in other mutant contexts (Albertson and Doe, 2003; Barros et al., 2003; Rolls et al., 2003; Slack et al., 2007). This may be irrelevant, but it may suggest that deletion of the BH motif affects other aspects of Mira behaviour and not only plasma membrane binding. Prompted by the reviewer’s comment, we have added this to the Discussion.

Regarding aPKC∆N overexpression, we agree that this could be a great test, however, it is very difficult to predict the precise consequences of overexpressing a deregulated kinase. Nonetheless, we found that Mira localization is telophase rescued (Peng et al., 2000) when aPKC∆N is overexpressed. We observed this by antibody staining against Mira and in living NBs monitoring Mira::mCherry localization. We have added this information into Figure 4—figure supplement 1’. These results suggest that loss of the BH motif and the effect of expressing constitutively active aPKC are not exactly the same as Mira retains the ability to engage with the cortex at least in telophase in the aPKC∆N context. aPKC∆N overexpression may bring about additional changes in the NBs cortex that prevent normal Mira polarization making it difficult to separate direct and indirect effects, which may be an alternative explanation why Mira is not localized directly after NEB in this context. We have added a section to the Discussion regarding this point.

Remarkably the authors don't even discuss the metaphase result in the paper! Finally, Miranda S96D (reduces but does not ablate BH interactions with the membrane) should have reduced cortical interactions in interphase but polarized normally in metaphase. Here again the result, reduced cortical signal in both interphase and metaphase, is not consistent with the author's proposed model.Remarkably, these results directly contradict the author's proposed model but are precisely as predicted from a simpler model in which aPKC regulated BH interactions with the membrane control Miranda cortical association throughout the cell cycle. Of course the more complex model could be saved by making it even more complex – perhaps the BH is also involved in the actomyosin interaction, perhaps aPKC also regulates the actomyosin anchor, perhaps a single point mutant that partially disrupts the BH lipid interaction also partially disrupts the actomyosin interaction – and the authors make a half-hearted attempt to do so (at least for the first experiment), but more data would need to be provided to support a more complex model (e.g. how aPKC might regulate the anchor).

We favor an alternative interpretation to the reviewer’s suggestion. The S96D mutation, by possibly bringing a negative charge to the BH motif, weakens the interaction the mutant protein with the plasma membrane. This has strong effects on S96D mutant localization in interphase, as it is cytoplasmic in the large majority of NBs analysed (Figure 3). This argues that in interphase Mira critically depends on plasma membrane binding for its cortical retention. In contrast, in all NBs analysed the S96D mutant after NEB shows cortical localization. This is perhaps weakened compared to controls, but occurs even with an asymmetric bias towards the basal pole. In this view, an additional mechanism that we propose to be mediated by actomyosin-dependent processes, overcomes the requirement for BH motif mediate maintenance of Mira after nuclear envelope breakdown. Therefore, these experiments are rather informative and support our idea that BH motif mediated retention of Mira after NEB is not the only mechanism that keeps Mira at the basal cortex.

I believe the problems outlined above represent deep and fundamental flaws and therefore preclude publication in its current form. I also note that the authors removed the model figure from the paper. In my opinion this is moving in the wrong direction and encourage the authors to revise the paper so it is more clear, not less.

We appreciate that the reviewer continues to provide constructive feedback, despite his/her contrary views on the matter. We agree that a graphical summary might improve reader friendliness and have added a model in new Figure 6. We focus our discussion also around the BH motif and have elaborated on our views, how actomyosin activity may contribute to Mira basal retention after nuclear envelope breakdown, although this remains to be determined in detail. This can be found in the Discussion section.

[Editors' note: further revisions were requested prior to acceptance, as described below.]

The core discovery of this manuscript 'Miranda localization is regulated differently during interphase and mitosis' is interesting and worth publishing. Knowing that Miranda localization switches the mode (plasma membrane-bound in interphase, and anchored via actin cytoskeleton in mitosis) is an important stepping stone to fully understand how Mira localization is regulated to achieve asymmetric neuroblast division.The reviewers had a lengthy discussion as to how to proceed with this manuscript. First, we agreed that the core discovery described above is worth publishing if accurately stated: that Miranda localization have two modes in interphase and mitosis. However, while several reviewers agreed that the data in the manuscript sufficiently supported their model, we were not able to reach a consensus on this point. In the end, the reviewers agreed to recommend publication provided that the authors clearly and accurately state their results in a manner that allows critical assessment of all the data, such that all the readers can judge on their own.

We mention the differences of Mira’s interaction with the cortex in the Abstract:

“We reveal a step-wise polarization of Miranda in larval neuroblasts and find that Miranda’s dynamics and cortical association are differently regulated between interphase and mitosis. In interphase Miranda binds to the plasma membrane. Then, before nuclear envelope breakdown, Miranda is phosphorylated by aPKC and displaced into the cytoplasm. This clearance is necessary for the subsequent establishment of asymmetric Miranda localization. After nuclear envelope breakdown, actomyosin activity is required to maintain Miranda asymmetry.”

We state that Miranda localization has two modes in interphase and mitosis in the last paragraph of the Introduction that summarizes our results:

“We reveal that Mira uses two modes to interact with the cortex: in interphase, to retain Mira uniformly at the cortex direct interaction of Mira’s BH motif with phospholipids of the PM are necessary and likely sufficient. This interaction is inhibited by aPKC-dependent phosphorylation of the BH motif at prophase. After nuclear envelope breakdown Mira requires BH motif and actomyosin dependent processes for asymmetric retention at the cortex. Therefore, we propose that Mira binds to the PM in interphase and to the actomyosin cortex in mitosis, both of which appear BH motif dependent.”

This is also stated in the third paragraph of the Discussion:

“We propose that Mira has two different modes by which it can be retained at the cortex (Figure 6). In interphase, Mira localizes uniformly to the cortex via direct interactions with the PM for which its BH motif is necessary and likely to be sufficient and which occurs independently of an intact F-actin cortex (Figure 2). After NEB, Mira still relies on the BH motif to localize in a basal crescent, but at this stage of the cell cycle actomyosin-dependent processes retain Mira basally (Figure 3, Figure 2). The transition between these localizations depends on phosphorylation by aPKC (Figure 3, Figure 4).”

All reviewers noted concerns regarding the accuracy of descriptions. More specific comments will follow below, but the collective result of inaccurate statements/descriptions is that 1) the real contribution of the manuscript is blurred,

We have added a new summary statement at the end of the Introduction, to better summarize our contribution:

“We reveal that Mira uses two modes to interact with the cortex: in interphase, to retain Mira uniformly at the cortex direct interaction of Mira’s BH motif with phospholipids of the PM are necessary and likely sufficient. This interaction is inhibited by aPKC-dependent phosphorylation of the BH motif at prophase. After nuclear envelope breakdown Mira requires BH motif and actomyosin dependent processes for asymmetric retention at the cortex. Therefore, we propose that Mira binds to the PM in interphase and to the actomyosin cortex in mitosis, both of which appear BH motif dependent.”

We also have changed the beginning of the Discussion to better summarize our approach and contribution:

“To address this apparent inconsistency, we reassessed in vivo the relative contribution of aPKC and actomyosin throughout the cell cycle analysing Mira localization using endogenously expressed reporters in living NBs. This has allowed us to resolve this problem as we find that asymmetric Mira localization is established stepwise and involves both aPKC-dependent phosphorylation and actomyosin-dependent anchoring, which are required at different time points in mitosis.

We propose that Mira has two different modes by which it can be retained at the cortex (Figure 6). In interphase, Mira localizes uniformly to the cortex via direct interactions with the PM for which its BH motif is necessary and likely to be sufficient and which occurs independently of an intact F-actin cortex (Figure 2). After NEB, Mira still relies on the BH motif to localize in a basal crescent, but at this stage of the cell cycle it might be required to mediate actomyosin-dependent basal retention of Mira (Figure 3, Figure 2). The transition between these localizations depends on phosphorylation by aPKC (Figure 3, Figure 4).”

2) The manuscript reads as if the authors are proposing an alternative model which they claim to be better than the existing model, but the new model also leaves many unexplained observations, making readers wonder whether the new model is a real improvement or only confusing the field.Therefore, we ask you to edit the text and submit the revised version. We decided to allow you one more round of revision because it will only require textual revision. With that said, all the reviewers expressed strong concerns that the current writing is not accurate enough, and if it is not fully taken care of in the next round, we must reject your manuscript, and no further revision will be allowed. In revising, we ask you to ensure that the revised main text is self-sufficient in conveying all the messages accurately and clearly, and not to utilize the rebuttal/response letter as a platform to explain things that are not entirely consistent with the main text.

We hope that the changes detailed above and below satisfactorily address your concerns.

Guidelines for revision:Each Results section should clearly indicate how the data contributes (or doesn't) to the model.

We have incorporated these changes into the main manuscript. We first introduced a statement to make clear what we test our results:

“According to this model, Mira retention at the cortex is primarily mediated through direct interaction with the PM mediated by its BH motif.”

The first section ends with the proposition that – based on the differences in intensities, Mira binding to the cortex might have occur through different binding modes:

“In conclusion, Miranda transitions from a uniformly cortical localization with low intensity levels in interphase, to a basal localization with high intensity levels in metaphase (Figure 1’, C). These cell-cycle dependent differences in cortical Mira intensities prompted the idea that Mira might use different modes of binding to the cortex in interphase versus mitosis. Therefore, we assayed for potential differences in cortical binding of Mira in interphase versus mitosis to address whether Mira is retained at the cortex primarily by BH motif interaction with the PM, or whether other modes of cortical retention contribute.”

The second section describing the results of LatA and ML-7 treatment, is in support of the model that Mira uses different modes to bind to the cortex in interphase and in mitosis. This can be found here):

“In summary, these results support the notion that Mira interacts differently with the cortex in interphase and in mitosis since F-actin and myosin activity contribute to establish and maintain asymmetric Mira crescents at the basal cortex following NEB, but they are not essential for uniform cortical localization of Mira in interphase nor for Mira clearance during prophase.”

The third section addressing the role of the BH motif describes data that support the model that Mira only uses one mode to bind to the cortex, regardless of the cell cycle stage, which is BH motif mediated interactions with the PM. This is stated here:

“These findings support the idea that, in interphase, the BH motif is necessary and likely to be sufficient to mediate interactions with phospholipids of the PM leading to uniform cortical localization of Mira. These findings also show that phospho-regulation of the BH motif affects Mira localization in both phases of the cell cycle. Therefore, these observations support the model that Mira uses only one mode, BH motif mediated PM interactions, for cortical association throughout the cell cycle. However, this model does not readily explain differences in the response of Mira to LatA and ML-7 in interphase versus mitosis (Figure 2).”

The fourth section describing the FRAP data supports the model that Mira has two different modes to interact with the cortex. This is stated:

“In conclusion, these results show that, in unperturbed NBs, Mira turnover at the PM in interphase and at the basal cortex in mitosis are different, supporting the notion that Mira has different binding modes in interphase versus mitosis. Furthermore, in apkc mutant NBs, instead of being cleared, Mira may persist throughout mitosis with the same actin-insensitive uniform localization, and similar turnover, as in interphase.”

The section detailing the effects of low concentrations of Y-27632 on Mira localization is compatible with Mira using two different modes for cortical retention in interphase and in mitosis. This statement can be found here:

“In conclusion, while Y-27632 at higher concentration, can indeed mimic the effect of apkc mutation on Mira, these results suggest that when NBs polarize in the presence of low concentrations of Y-27632, Mira crescent size is affected, which is likely to occur independently of aPKC inhibition. These results suggest that Mira cortical retention has different mechanisms of regulation in interphase and in mitosis. They also hint at an additional, Y-27632 sensitive layer of regulation controlling basal Mira crescent size.”

We also ask that in the discussion the authors recognize that there are observations they have made that are inconsistent with this model. They can do their best to try and explain them away, but at least readers will be able to more readily appreciate these inconsistencies and judge the author's explanations for themselves. Please ensure that the revised manuscript is self-sufficient in conveying these points, without the need of relying on response letter.

We have modified the Discussion highlighting the data we and other obtained that are consistent with the BH motif mediated PM interactions being the primary mode of Mira cortical retention throughout the cell cycle. This can be found in the Discussion:

“We observe that deletion of the BH motif as well as overexpression of aPKC^ΔN^ disrupt cortical localization of Mira in interphase and mitosis (Figure 3 and Figure 3—figure supplement 1) and that the phosphomimetic S96D mutation reduces Mira localization in interphase as well as in mitosis (Figure 3). These findings argue for the model that throughout the cell cycle Mira cortical association depends solely on BH motif mediated interaction with the PM, that is negatively regulated by locally controlled aPKC phosphorylation (Atwood et al., 2009; Bailey et al., 2015). “

and: “What could be the role of F-Actin for Mira localization in this model? F-Actin clearly contributes to aPKC regulation of Miranda localization by restricting the localization of the Par complex to the apical pole as LatA addition changes the distribution of aPKC and Baz (Figure 2).”

as well as: *“We propose that the BH motif may mediate Mira’s interaction with actomyosin, which remains to be tested.”*

The reviewers consider that the most reasonable model to be put forward in this manuscript is "either Miranda BH-phospholipid or BH-actomyosin interactions are sufficient for cortical localization. In interphase BH-phospholipid interactions mediate uniform cortical association but early in prophase they start becoming inactivated by aPKC in an apical to basal fashion. By metaphase all BH-phospholipid interactions are inactivated and BH-actomyosin interactions take over (BH-phospholipid interaction might have supportive role in mitosis, too).

We have followed the advice of the reviewers and discuss this possibility. This can be found in the Discussion:

“We propose that Mira has two different modes by which it can be retained at the cortex (Figure 6). In interphase, Mira localizes uniformly to the cortex via direct interactions with the PM for which its BH motif is necessary and likely to be sufficient and which occurs independently of an intact F-actin cortex (Figure 2). After NEB, Mira still relies on the BH motif to localize in a basal crescent, but at this stage of the cell cycle it might be required to mediated actomyosin-dependent processes Mira basal retention (Figure 3, Figure 2). The transition between these localizations depends on phosphorylation by aPKC (Figure 3, Figure 4).”

and in the new Figure 6 and in its legend:

“Figure 6. Model. Mira associates with the cortex using two different modes, which characteristics are detailed in the bottom row. During interphase, Mira directly binds to the phospholipids of the PM via its BH motif (black double arrow). During prophase, aPKC-dependent phosphorylation of this motif abolishes this interaction, resulting in the progressive clearance of Mira from the cortex, in an apical-to-basal manner driven Mira into the cytoplasm. This clearance in prophase is necessary for Mira to associate with the basal cortex after NEB, via Actomyosin-dependent retention. Both the precise phosphoregulation and molecular characteristics of this mode remain to be determined. The BH motif, also required at this step, may directly or indirectly mediate interactions between Mira and actomyosin (green double arrow). PM interactions via its BH motif (black double arrow) may still contribute, but are not sufficient to mediate Mira basal retention after NEB.”

Overall, the reviewers agreed that there are many observations that do not exactly fit to the authors' model (e.g. overexpression of aPKC-deltaN). Some experiments were suggested to test authors' model, which resulted in observations that are inconsistent with the authors model. Each time, the authors provided 'possible explanations why the test did not support their model', yet remained that 'the model is still correct'. The authors should put more effort into explaining these inconsistent results (instead of 'explaining away' inconvenient results, trying to reach better explanation that makes sense as a whole). The reviewers do not expect the authors to figure out the mechanism of Miranda localization entirely. The reviewers appreciate that knowing that Miranda localization likely has two modes is sufficiently important progress, but writing is not conveying that this is the core message of the manuscript, and instead it claims more than the experimental data can support.

As mentioned above we have revised the discussion according to the reviewer’s advice.

We have modified the discussion highlighting the data we and other obtained that are consistent with the BH motif being the primary mode of Mira cortical retention throughout the cell cycle. This can be found in the Discussion:

“We observe that deletion of the BH motif as well as overexpression of aPKC^ΔN^ disrupt cortical localization of Mira in interphase and mitosis (Figure 3 and Figure 3—figure supplement 1) and that the phosphomimetic S96D mutation reduces Mira localization in interphase as well as in mitosis (Figure 3). These findings by themselves argue for the model that throughout the cell cycle Mira cortical association depends solely on BH motif mediated interaction with the PM, that is negatively regulated by locally controlled aPKC phosphorylation (Atwood et al., 2009; Bailey et al., 2015). “

and:

“What could be the role of F-Actin for Mira localization in this model? F-Actin clearly contributes to aPKC regulation of Miranda localization by restricting the localization of the Par complex to the apical pole as LatA addition changes the distribution of aPKC and Baz (Figure 2).”

As an example of inaccurate statement, in the first section of the results, the authors start out by stating Miranda is 'cleared' and 'reappears' in mitosis, clearly leaving the impression that the previous model of expanding apical Par complex gradually displacing Miranda is wrong. However, as the video shows, Miranda 'clearance' occurs from the apical to basal. Although the authors explain that this clearing happens from apical to basal, but in subsequent sections, they do not come back to this fact and stick to the expression of 'clearance' and 'reappearance'. Here, the emphasis should be the fact that mitotic Miranda crescent is much stronger, prompting the investigation of the mechanism by which Miranda anchoring is promoted during mitosis (which they show to be actin dependent).

We follow the advice of the reviewers and use differences in Mira cortical intensities as an argument to further investigate if differences in Mira binding to the cortex in interphase and mitosis exist. This can be found in the Results section:

“In conclusion, Miranda transitions from a uniformly cortical localization with low intensity levels in interphase, to a basal localization with high intensity levels in metaphase (Figure 1’, C). These cell-cycle dependent differences in cortical Mira intensities prompted the idea that Mira might use different modes of binding to the cortex in interphase versus mitosis. Therefore, we assayed for potential differences in cortical binding of Mira in interphase versus mitosis.”

The weakest explanations in the current manuscript are the following:1) If the authors' model is entirely correct, aPKC-deltaN expression should result in delocalization of Miranda in interphase, but normal, crescent localization in mitosis. But the actual observation is that aPKC-deltaN results in the delocalization of Miranda in interphase and mitosis. In the main text, the authors do not mention that Miranda fails to localize even in mitosis upon aPKC-deltaN expression, and conclude that their data is consistent with their model. In explaining this inconsistency, they state 'it is very difficult to predict the precise consequences of overexpressing a deregulated kinase'. A better explanation is required for the authors to be able to propose that their new model fits better than the existing model with all experimental results.

We mention now in the main text that aPKC-deltaN results in the delocalization of Miranda in interphase and mitosis. This can be found in the Results section:

“Finally, ectopic activation of aPKC by overexpression of constitutively active aPKC^ΔN^ (Betschinger et al., 2003b) also prevented Mira cortical localization in interphase and most of mitosis.”

We also provide a better explanation. i.e. that control of the levels of Mira phosphorylation might be important, which can be found in the Discussion:

“We observe that deletion of the BH motif as well as overexpression of aPKC^ΔN^ disrupt cortical localization of Mira in interphase and mitosis (Figure 3 and Figure 3—figure supplement 1) and that the phosphomimetic S96D mutation reduces Mira localization in interphase as well as in mitosis (Figure 3). These findings argue for the model that throughout the cell cycle Mira cortical association depends solely on BH motif mediated interaction with the PM, that is negatively regulated by locally controlled aPKC phosphorylation (Atwood et al., 2009; Bailey et al., 2015).”

MiraS96D: the model predicts that MiraS96D should be cortically polarized in metaphase but it's partially cytoplasmic (S96D only partially destabilizes the BH-phospholipid interaction). The author's explanation is that cortical localization is affected more in interphase, which is not a thorough discussion.

We discuss now two possibilities. The S96D results are consistent with the model that BH motif mediated interactions with the plasma membrane retaining Mira at the cortex in interphase and mitosis, as Mira localization is reduced in both cases. This can be found in the Discussion:

“We observe that deletion of the BH motif as well as overexpression of aPKC^ΔN^ disrupt cortical localization of Mira in interphase and mitosis (Figure 3 and Figure 3—figure supplement 1) and that the phosphomimetic S96D mutation reduces Mira localization in interphase as well as in mitosis (Figure 3). These findings argue for the model that throughout the cell cycle Mira cortical association depends solely on BH motif mediated interaction with the PM, that is negatively regulated by locally controlled aPKC phosphorylation (Atwood et al., 2009; Bailey et al., 2015).”

However, this does not take into account the observation that the S96D mutation affects interphase localization of Mira stronger than its mitotic localization (In ~70% of interphase NBs S96D mutant Mira does not localize to the cortex, while S96D mutant Mira always localizes to the cortex of mitotic NBs). These results argue that the effect of the Aspartate on Mira localization is less strong in mitosis, which is the basis for the interpretation that basal Mira is stabilized by additional interactions after NEB.

This can be found in the Discussion:

“The phosphomimetic S96D mutation strongly disrupts uniform localization to the PM in interphase, but localizes at the basal cortex in mitosis, albeit at reduced levels (Figure 3). This is consistent with the existence of Mira stabilizing interactions in mitosis, that are not present in interphase, which reduce the effect of a negative charge provided by the Aspartate in the phosphomimetic mutant on cortical Mira localization.”

Unfortunately, readers will be wondering why Miranda5D (which completely abolishes BH-phospholipid interactions) wasn't tested because it would very easily resolve the explanations provided by the author for both of these inconsistencies (especially because the explanations are so poor).

We agree that this would be interesting to test. In this context, it would be important to determine which sites in Mira are phosphorylated in vivo, too (e.g. pulling down endogenous Mira from control and aPKC mutant samples followed by mass spectrometry approaches).

2) According to the authors' model, Mira-deltaBH should be cytoplasmic in interphase, yet should localize to the basal crescent in mitosis. However, they find that Mira-deltaBH fails to localize to the crescent even in mitosis. In the main text, they simply conclude that 'this is consistent with Mira being localized to the cortex via interaction with plasma membrane'. Then, in response to the reviewers' comment that deltaBH shouldn't be cytoplasmic in mitosis, only in response letter, they point out that mitotic Miranda localization may require BH domain-mediated plasma membrane localization in addition to actomyosin. The authors should avoid any discrepancies between the main text and the response letter, and a cohesive story must be presented within the main text (and the observations that do not fit to the model must be clearly presented and acknowledged).

We have updated our model following the reviewer’s advice stating that the BH motif may mediate also interaction with actomyosin. This statement can be found in our summarizing statement of the Introduction:

“Therefore, we propose that Mira binds to the PM in interphase and to the actomyosin cortex in mitosis, both of which appear BH motif dependent.”

This statement is further found in the Discussion:

“A surprising finding is that the BH motif is essential for Mira localization in interphase and in mitosis. In mitosis, BH motif mediated PM binding is no longer sufficient to localize Miranda to the basal pole. This is indicated by the requirement of the actomyosin cytoskeleton after NEB (Figure 2). However, the BH motif is still necessary for Mira localization after NEB. It is possible that the BH-phospholipid interactions still play a role in mitosis. Deletion of the BH motif could also cause more indirect effects. For example, Mira∆BH is not found on mitotic microtubules, where Mira is typically observed in conditions where it is unable to localize correctly (Albertson and Doe, 2003; Barros et al., 2003; Rolls et al., 2003; Slack et al., 2007). We propose that the BH motif may mediate Mira’s interaction with actomyosin, which this remains to be tested.”

Also, the reviewers do not agree that the data shows that Miranda disappears all at once upon LatA treatment. The revision should remove this argument and focus on the argument that Miranda disappears before aPKC arrives. Of course, we do not ask the authors to figure out everything about Mira localization, but they should explain 'why a particular observation betrays their prediction based on the model' in a cohesive manner, and more sincerely (i.e. do not bury important discussions within the response letter, which become inconsistent when placed in the main text).

We have revised this Results section and removed the data, that we feel shows that Mira levels decline at similar rates at the extremities of the crescent and in its center. We now use the timing of changes in Mira localization versus changes in Par complex distribution in response to LatA as an argument as advised by the reviewers. This can be found in the Results section:

“However, LatA treatment after NEB also led to the redistribution of Baz/Par3 and aPKC to the entire NB periphery. Therefore, the observed effect on Mira could be indirect, caused by changes in aPKC localization when F-actin is compromised. Assuming that aPKC activity is restricted to the cortex (Atwood et al., 2007; Rodriguez et al., 2017) we sought to distinguish between direct and indirect effects on Mira by determining if Mira loss preceded (indicative of direct effect of loss of actin) or followed (indicative of an indirect effect caused by changes in aPKC localization) changes in Par complex distribution in colcemid arrested NBs upon LatA treatment. We found that Mira loss preceded changes in cortical aPKC/Baz localization in response to LatA. This occurred about 2.8 ± 1min (n=13) before aPKC (Video S4) or Baz (Video S5) became detectable at the basal cortex (Figure 2—figure supplement 1).”

We would like to provide an example of how reviewers' discussion proceeded. One reviewer noted: "I prefer an alternative interpretation, in which all of the localization after NEB is actomyosin dependent (ML-7 abolishes the crescent) and that this is mediated by the BH domain.": this clearly suggests that this reviewer appreciated the authors' discovery but still required to introduce his/her own interpretation to support authors' model. In response, another reviewer asked, "Then why is Miranda cortical in metaphase neuroblasts treated with LatA and lacking aPKC function (Figure 4—figure supplement 1 panel B)? If localization after NEB is actomyosin dependent, then surely it shouldn't localize after treatment with LatA." In response to this question, the first reviewer responded, "My interpretation of this experiment is that it indicates that aPKC-dependent clearing is an essential prerequisite for the formation of the basal crescent. In the absence of aPKC, Miranda remains in the interphase state where it binds to phospholipids. I could be wrong, as this result is entirely consistent with the simple (existing) model, but the latter doesn't explain the ML-7, Y26732 and FRAP data. What I find harder to understand is why the basal crescent doesn't form in the presence of constitutively-active aPKC, which is why the results aren't clear cut. It seems that Miranda needs to be phosphorylated by aPKC before NEB to form the basal crescent, but that too much aPKC activity or activity after NEB inhibits this. It is a shame that the manuscript didn't really get to grips with these questions."In our opinion, a manuscript should not leave this much room of interpretation to the readers, and the authors must clearly present their data (or if that is not sufficient, more data would be clearly required. But the reviewers are not asking more experiments here).

We feel that the advice to clarify what is tested (PM binding explains Mira cortical localization throughout the cell cycle or whether different binding modes exist) and to state at the end of each Results section as well as in the Discussion which data fit one primary binding mode and which support different modes has helped to address these points. (See above).

The Abstract states:*”*This clearance is necessary for the subsequent establishment of asymmetric Miranda localization.”

We discuss the possibility that aPKC phosphorylation at prophase is important for basal Mira crescent formation after NEB:

“This aPKC-dependent step in prophase might be a prerequisite for basal crescent formation in metaphase. One possibility is that phosphorylation of the BH motif might potentiate Mira’s ability to engage with actomyosin for basal retention after NEB. Mira phosphorylation might need to be properly balanced and locally controlled to allow for Mira asymmetric localization. This could explain why the phosphomimetic S96D mutant, displays reduced basal localization in metaphase and that upon overexpression of aPKC^ΔN^ Mira does not form basal crescents in metaphase (Figure 3—figure supplement 1).”

Additionally, the model figure is currently extremely vague because it is hard to see what the authors are implying is taking place as far as Miranda-cortex interactions. Please revise the model figure to describe your model more clearly.

We have revised Figure 6 to include more graphical details and added a legend describing that Mira has two modes to bind to the cortex that are different in interphase and in mitosis. The new legend can be found here:

“Figure 6. Model. Mira associates with the cortex using two different modes, which characteristics are detailed in the bottom row. During interphase, Mira directly binds to the phospholipids of the PM via its BH motif (black double arrow). During prophase, aPKC-dependent phosphorylation of this motif abolishes this interaction, resulting in the progressive clearance of Mira from the cortex, in an apical-to-basal manner driven Mira into the cytoplasm. This clearance in prophase is necessary for Mira to associate with the basal cortex after NEB, via Actomyosin-dependent retention. Both the precise phosphoregulation and molecular characteristics of this mode remain to be determined. The BH motif, also required at this step, may directly or indirectly mediate interactions between Mira and actomyosin (green double arrow). PM interactions via its BH motif (black double arrow) may still contribute, but are not sufficient to mediate Mira basal retention after NEB.”